# Pervasive transcription fine-tunes replication origin activity

Tito Candelli[†‡], Julien Gros[†]*, Domenico Libri*

Institut Jacques Monod, CNRS UMR 7592, Université Paris Diderot, Sorbonne Paris Cité, Paris, France

**Abstract** RNA polymerase (RNAPII) transcription occurs pervasively, raising the important question of its functional impact on other DNA-associated processes, including replication. In budding yeast, replication originates from Autonomously Replicating Sequences (ARSs), generally located in intergenic regions. The influence of transcription on ARSs function has been studied for decades, but these earlier studies have neglected the role of non-annotated transcription. We studied the relationships between pervasive transcription and replication origin activity using high-resolution transcription maps. We show that ARSs alter the pervasive transcription landscape by pausing and terminating neighboring RNAPII transcription, thus limiting the occurrence of pervasive transcription within origins. We propose that quasi-symmetrical binding of the ORC complex to ARS borders and/or pre-RC formation are responsible for pausing and termination. We show that low, physiological levels of pervasive transcription impact the function of replication origins. Overall, our results have important implications for understanding the impact of genomic location on origin function.

DOI: https://doi.org/10.7554/eLife.40802.001

*For correspondence:
julien.gros@ijm.fr (JG);
domenico.libri@ijm.fr (DL)

[†]These authors contributed equally to this work

Present address: [‡]Princess Máxima Center for Pediatric Oncology, Utrecht, The Netherlands

Competing interests: The authors declare that no competing interests exist.

## Introduction

The annotation of transcription units has traditionally heavily relied on the detection of RNA molecules. However, in the last decade, many genome-wide studies based on the direct detection of RNA polymerase II (RNAPII) have clearly established that transcription extends largely beyond the limits of regions annotated for coding functional RNA or protein products (*Jacquier, 2009*; *Porrua and Libri, 2015*). The generalized presence of transcribing RNA polymerases, not necessarily associated to the production of stable RNAs, defines pervasive or hidden transcription, which is a conserved feature of both eukaryotic and prokaryotic transcriptomes.

In *S. cerevisiae*, pervasive transcription accounts for the production of a multitude of transcripts generally non-coding, many of which undergo degradation in the nucleus or the cytoplasm (*Jacquier, 2009*; *Porrua and Libri, 2015*). Transcription termination limits the extension of many non-coding transcription events, compensating, to some extent, the promiscuity of initiation (for recent reviews see: *Jensen et al., 2013*; *Porrua and Libri, 2015*). In *Saccharomyces cerevisiae* cells, two main pathways are known for terminating normal and pervasive RNAPII transcription events (*Porrua et al., 2016*). The first is employed for termination of mRNA coding genes and depends on the CPF-CF (cleavage and polyadenylation factor-cleavage factor) complex. Besides participating in the production of mRNAs, this pathway is also important for transcription termination of several classes of non-coding RNAs, namely SUTs (stable unannotated transcripts) and XUTs (Xrn1-dependent unstable transcripts) (*Marquardt et al., 2011*). Transcription terminated by this pathway produces RNAs that are exported to the cytoplasm and enter translation. If they contain premature stop codons, they are subject to the nonsense mediated decay and might not be detected in wild-type cells (*van Dijk et al., 2011*; *Malabat et al., 2015*).

The second pathway depends on the NNS (Nrd1-Nab3-Sen1) complex and is responsible for terminating transcription of genes that do not code for proteins. Small nucleolar RNAs (snoRNAs) and cryptic unstable transcripts (CUTs), a prominent class of RNAPII pervasive transcripts, are typical targets of NNS-dependent termination. One important feature of this pathway is its association with proteins involved in nuclear RNA degradation such as the exosome and its cofactor, the Trf4-Mtr4-Air (TRAMP) complex. The released RNA is not exported to the cytoplasm but polyadenylated by TRAMP and nucleolytically attacked by the exosome that trims snoRNAs to their mature length and fully degrades CUTs.

Recent studies in yeast and other eukaryotes have shown that constitutive and regulated read-through at terminators provides a very significant contribution to pervasive transcription (*Vilborg et al., 2015*; *Grosso et al., 2015*; *Rutkowski et al., 2015*; *Candelli et al., 2018*). Fail-safe mechanisms are in place to back up termination and restrict transcription leakage at terminators. One of these mechanisms terminates 'stray' transcription by harnessing the capability of DNA-bound proteins to roadblock RNAPII. Roadblocked polymerases are then released from the DNA via their ubiquitination and likely degradation (*Colin et al., 2014*).

The ubiquitous average coverage of the genome by transcription, coupled to the remarkable stability of the transcription elongation complex, raises the important question of the efficient coordination of machineries that must read, replicate, repair and maintain the same genomic sequences. The crosstalks between transcription and replication are paradigmatic in this respect.

Eukaryotic cells faithfully duplicate each of their chromosomes by initiating their replication from many origin sites (*Bell and Labib, 2016*). To ensure once-and-only-once DNA replication per cell cycle, coordination of initiation from these different sites is guaranteed by a two-step mechanism: replication origins have to be licensed before getting activated (*Diffley, 2004*). Licensing occurs from late mitosis to the end of G1 and consists in the deposition of pre-RCs (pre-replication complexes) around origin sites. To do so, ORC (origin recognition complex) recognizes and binds specifically origin DNA where it recruits Cdc6 and Cdt1 to coordinate the deposition of the replicative helicase engine, the hexameric Mcm2-7 complex. At each licensed origin is deposited a pair of Mcm2-7 hexamers assembled head-to-head as a still inactive double-hexamer (DH) encircling DNA. At the G1/S transition and throughout S-phase, the orderly recruitment of firing factors onto the Mcm2-7 DH activates it, ultimately triggering the building of two replisomes synthesizing DNA from the origin (*Parker et al., 2017*).

*S. cerevisiae* origins are specified in cis by the presence of Autonomously Replicating Sequences (ARSs). Within each ARS, ORC recognizes and binds specifically a bipartite DNA sequence composed of the ACS (ARS Consensus Sequence, 5'-WTTTATRTTTW-3'; *Palzkill and Newlon, 1988*; *Diffley and Cocker, 1992*; *Bell and Stillman, 1992*) and the B1 element (*Rao and Stillman, 1995*; *Li et al., 2018*). The ACS oriented by its T-rich strand is generally found at the 5' ends of ARS sequences (*Eaton et al., 2010*). A-rich stretches are often present at the opposite end of ARSs and have been proposed to function as additional ACSs oriented opposite to the main ACS (*Breier et al., 2004*; *Yardimci and Walter, 2014*). Such secondary ACSs have been shown to strengthen pre-RC assembly at ARS *in vitro* and proposed to ensure ARS function *in vivo* by driving the cooperative recruitment of a second ORC (*Coster and Diffley, 2017*; see also *Warner et al., 2017*). This contrasts with earlier *in vitro* reconstitutions of pre-RC assembly on single DNA molecules, supporting the recruitment of only one ORC per DNA (*Ticau et al., 2015*; *Duzdevich et al., 2015*). Whether one or two ORC molecules are recruited at ARSs *in vivo* for efficient pre-RC assembly is still not fully understood.

ACS presence is necessary but not sufficient for ARS function *in vivo*, as only a small fraction of all ACSs found in the *S. cerevisiae* genome corresponds to active ARSs (*Tuduri et al., 2010*). Other DNA sequence elements and factors, including the structure of chromatin, participate to origin specification and usage. On the one hand, ORC binding at the ACS shapes NFR formation, nucleosome positioning and nucleosome occupancy, which all together maximize pre-RC formation (*Lipford and Bell, 2001*; *Eaton et al., 2010*; *Belsky et al., 2015*; *Rodriguez et al., 2017*). On the other hand, specific histone modifications mark replication initiation sites (*Unnikrishnan et al., 2010*) and chromatin-coupled activities ensure replication forks progression and origin efficiency (*Kurat et al., 2017*; *Devbhandari et al., 2017*; *Azmi et al., 2017*). The transcription machinery could participate to the establishment of a specific chromatin landscape and/or play a more direct role in the

specification and function of origins. However, to what extent annotated and non-annotated transcription at and around origins can influence replication remains unclear.

The binding of general transcription factors such as Abf1 and Rap1, or even the tethering of transcription activation domains, TBP or Mediator components was shown to be required for efficient firing of a model ARS (*Marahrens and Stillman, 1992*; *Stagljar et al., 1999*; see also *Knott et al., 2012*). However, whether this implies the activation of transcription within origins has not been shown.

Strong transcription through ARSs has been demonstrated to be detrimental for their function (*Snyder et al., 1988*; *Tanaka et al., 1994*; *Chen et al., 1996*; *Mori and Shirahige, 2007*; *Lõoke et al., 2010*), and intragenic origins have been shown to be inactivated by meiotic-specific transcription (*Mori and Shirahige, 2007*; *Blitzblau et al., 2012*). Inactivation of origins by transcription has been correlated to the impairment of ORC binding and pre-RC assembly, possibly because of steric conflicts with transcribing RNAPII (*Mori and Shirahige, 2007*; *Lõoke et al., 2010*). Strong transcription through origins was found to terminate, at least to some extent, within ARS sequences at cryptic termination sites, generating stable and polyadenylated transcripts (*Chen et al., 1996*; *Magrath et al., 1998*). However, it was concluded that transcription termination within ARSs and origin function are not functionally linked, as mutationally impairing either one would not affect the other. In particular, it was found that transcription termination was not due to ORC roadblocking RNAPII and, conversely, that origin activity was not dependent on termination taking place within the ARS (*Chen et al., 1996*; *Magrath et al., 1998*).

Even if unrestricted transcription inactivates intragenic origins (*Mori and Shirahige, 2007*; *Blitzblau et al., 2012*), these cases hardly represent the chromosomal context of most mitotically active origins, which are intergenic (*Donato et al., 2006*; *MacAlpine and Bell, 2005*; *Nieduszynski et al., 2005*) and are generally not exposed to the levels of transcription found within genes. Most importantly, these earlier studies could not take into account the potential impact of annotated and non-annotated levels of pervasive transcription, which is not easily detected, due to the general instability of the RNA produced and to the poor resolution of many techniques for detecting RNAPII occupancy. Such generally low levels of transcription have been recently found to significantly impact the expression of canonical genes and to be limited by fail safe and redundant transcription termination pathways (*Candelli et al., 2018*; *Roy et al., 2016*).

We investigated here the impact of physiological levels of pervasive transcription on the function of replication origins in *S. cerevisiae*. Using nucleotide-resolution transcription maps, we studied the transcriptional landscape around and within origins, regardless of annotations. Origins generate a characteristic footprint in the ubiquitous transcriptional landscape due to the pausing of RNAPII at origin borders. On the one hand, transcription terminates at the border of the primary ACS, in an ORC and pre-RC-dependent manner, by a mechanism that has roadblock features. On the other hand, RNAPII pauses upstream of the secondary ACS but terminates within the ARS. The low levels of pervasive transcription that enter ARSs negatively affect the efficiency of licensing and firing, with pervasive transcription incoming from the secondary ACS affecting origin function to a higher extent.

These results have important implications for understanding the impact of genomic location on origin specification, efficiency and timing of activation. Because pervasive transcription is conserved and generally increases with increased genome complexity, they are also susceptible to be relevant for the mechanism of replication initiation in other eukaryotes, particularly in metazoans.

## Results

### RNAPII pausing and transcription termination occur at ARS borders

Although considerable efforts have been made to annotate transcription units independently from the production of stable RNAs, many transcribed regions still remain imprecisely or poorly annotated in the *S. cerevisiae* genome. Addressing the potential impact of transcription on the function of replication origins therefore requires taking into account the actual physiological levels of transcription, regardless of annotation. For these reasons, we relied on high-resolution transcription maps derived from the direct detection of RNAPII by the sequencing of the nascent transcript (RNAPII PAR-CLIP, photo-activable ribonucleoside-enhanced UV-crosslink and immunoprecipitation)

(*Schaughency et al., 2014*). We also generated additional datasets using the analogous RNAPII CRAC, (crosslinking analysis of cDNAs, *Granneman et al., 2009*; *Candelli et al., 2018*). Both methods detect significant levels of transcription in many regions that lack annotations (data not shown; *Candelli et al., 2018*).

We retrieved a total of 228 origins that we oriented according to the direction of the T-rich strand of their proposed ACS (*Nieduszynski et al., 2006*). Origins were then anchored at the 5' ends of their ACS and the median distribution of RNAPII occupancy was plotted in a 1 kb window around the anchoring site (*Figure 1A*). Strikingly, RNAPII signal accumulates over the 200nt preceding the T-rich strand of the ACS and sharply decreases within the 25nt immediately preceding it (*Figure 1A*, blue trace; see also *Figure 1—figure supplement 2A–B* for the statistical significance of the signal loss over the primary ACSs). The RNAPII signal build-up suggests that pausing occurs before the ACS, while its abrupt reduction might indicate that transcription termination occurs immediately upstream of the site. This behavior is reminiscent of roadblock termination whereby transcription elongation is impeded by factors or complexes binding the DNA, and RNA polymerase is released following its ubiquitylation (*Colin et al., 2014*; *Roy et al., 2016*; *Candelli et al., 2018*). RNAPII signal also builds up from antisense transcription, although in a more articulated manner (*Figure 1A*, red trace) and starts declining on average 120nt upstream of the 5' border of the ACS.

Although the sharp decrease of RNAPII signal immediately preceding the ACS is suggestive of transcription termination, it is possible that RNAPII occupancy downstream of the ACS decreases because of a shorter persistency of the elongation complex in these regions, for instance because of higher transcription speed. We thus sought independent evidence of transcription termination before the ACS. Transcription termination is accompanied by release of the transcript and generally by its polyadenylation. Therefore, we mapped the distribution of polyadenylated RNA 3'-ends around origins as a proxy for transcription termination (*Figure 1B*, blue). Because roadblock termination produces RNAs that are mainly degraded in the nucleus, we also profiled the distribution of RNA 3'-ends in cells depleted for the two catalytic subunits of the exosome, Rrp6 and Dis3 (*Roy et al., 2016*) (*Figure 1B*, transparent red). At each position around the ACS, we scored the number of genomic sites containing at least one RNA 3'-end without taking into consideration the read count at each site. This conservative strategy determines whether termination occurs at each position, and prevents high read count values from dominating the aggregate value. The distribution of RNA 3'-ends – and therefore of transcription termination events – closely mirrors the distribution of RNAPII on the T-rich strand of the ACS and peaks immediately upstream of the ACS. Note that because the whole read is taken into account to map RNAPII distribution, while only the terminal nucleotide is used to map the 3'-ends, the distribution of RNA 3'-ends is shifted downstream relative to the distribution of RNAPII. Importantly, and consistent with a roadblock mechanism, the 3'-end count upstream of the ACS is higher in the absence of the exosome (*Figure 1B*, transparent red), strongly suggesting that these termination events produce, at least to some extent, RNAs that are degraded in the nucleus. These peaks of RNA 3'-ends are significant, as demonstrated by the p-values associated to the frequencies of termination events observed around the ACS, which are significantly smaller than the ones detected in the flanking region (corrected p-value$<10^{-20}$, *Figure 1—figure supplement 2D* and Material and methods).

These observations strongly suggest that the landscape of pervasive transcription is significantly altered by the presence of replication origins. Incoming RNAPIIs are paused with an asymmetric pattern around ARSs and termination occurs upstream of the primary ACS.

To assess the origin of the asymmetry in RNAPII distribution, we considered the possibility that RNAPIIs transcribing in the antisense direction relative to the ACS might be paused at the level of putative secondary ACSs located downstream within the ARS. Such secondary ACSs, proposed to be positioned 70-400nt downstream and in the opposite orientation of the main ACS, have been shown to be required *in vitro* for efficient pre-RC assembly and suggested to play an important role for origin function *in vivo* (*Coster and Diffley, 2017*). The variable position of these secondary ACS sequences could explain why the antisense RNAPII meta-signal spreads over a larger region when ARSs are aligned to the 5' ends of their primary ACSs (*Figure 1C*). We therefore mapped such putative secondary ACSs using a consensus matrix derived from the set of known primary ACSs (*Coster and Diffley, 2017*) (Table 2). As shown on *Figure 1—figure supplement 1A*, distances between the primary and the predicted secondary ACS distribute widely and preferentially cluster around ≈100nt (median 113.5), consistent with functional data obtained using artificial constructs

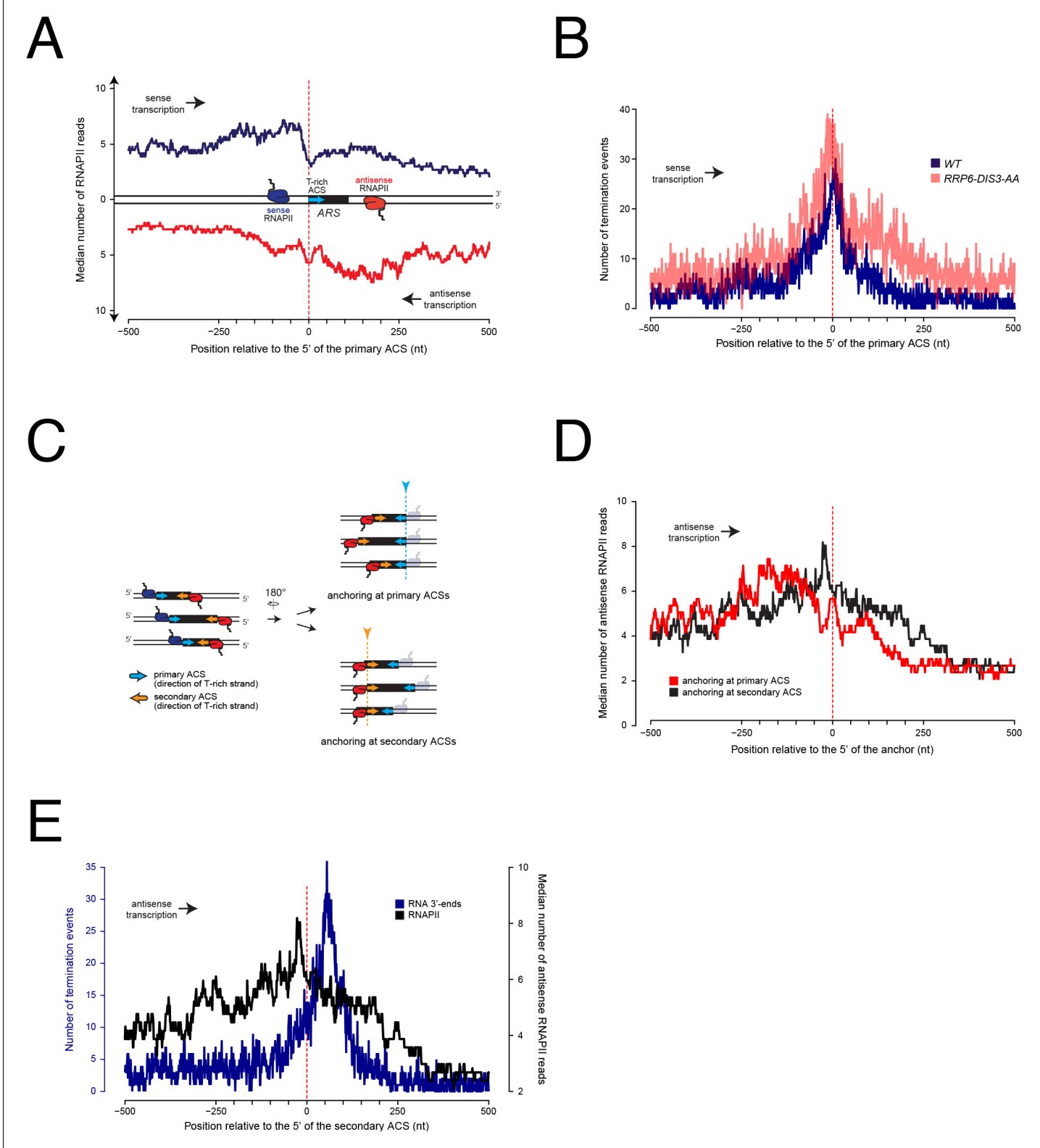

**Figure 1.** Metasite analysis of RNAPII occupancy and transcription termination at replication origins. (**A**) RNAPII PAR-CLIP metaprofile at replication origins. 228 confirmed ARSs were oriented according to the direction of the T-rich strand of their proposed ACSs (blue arrow) (*Nieduszynski et al.,* *2006*) and aligned at the 5' ends of the oriented ACSs (red dashed line). The median number of RNAPII reads (*Schaughency et al., 2014*) calculated for each position is plotted. Transcription proceeding along the T-rich strand of the ACS is represented in blue and considered to be sense, while

*Figure 1 continued on next page*

*Figure 1 continued*

transcription on the opposite strand is plotted in red and considered to be antisense. (B). Distribution of poly(A)+RNA 3'-ends at genomic regions surrounding replication origins. Origins were oriented and anchored as in A). 3'-ends reads (*Roy et al., 2016*) of RNAs extracted from wild-type cells (WT, blue) or cells in which both Rrp6 and Dis3 were depleted from the nucleus (*RRP6-DIS3-AA*, transparent red) were plotted. At each position around the anchor, the presence or absence of an RNA 3'-end was scored independently of the read count. (C). Scheme of replication origins anchored at different ACS sequences. Left: sense polymerases transcribing upstream of primary ACSs (blue arrows) are colored in blue, while antisense polymerases transcribing upstream of secondary ACSs (orange arrows) are colored in red. Right: ARSs oriented according to antisense transcription were aligned at the 5' ends of the primary ACSs (top, corresponds to red trace in D) or at the 5' ends of the secondary ACSs (bottom, corresponds to black trace in D). (D). RNAPII PAR-CLIP metaprofile of antisense transcription aligned either to the 5' ends of the primary (red) or the secondary (black) ACSs, as shown in (C). As in (A), the median number of RNAPII reads calculated for each position is plotted. (E). Distributions of RNA 3'-ends and RNAPII at genomic regions aligned at secondary ACSs. Origins were oriented and aligned as in (D). At each position around the anchor, presence or absence of an RNA 3'-end was scored independently of the read count (left y-axis). The distribution of RNAPII already shown in (C) is reported here for comparison (right y-axis).

DOI: https://doi.org/10.7554/eLife.40802.002

The following figure supplements are available for figure 1:

**Figure supplement 1.** Measures on mapped secondary ACSs.

DOI: https://doi.org/10.7554/eLife.40802.003

**Figure supplement 2.** Statistical analysis of pausing and termination signals.

DOI: https://doi.org/10.7554/eLife.40802.004

(*Coster and Diffley, 2017*). As possibly expected, the calculated similarity scores for these predicted ACSs are generally lower than the ones calculated for the main ACSs (see the distribution in *Figure 1—figure supplement 1B*). When we aligned origins to the first position of their predicted secondary ACSs (*Figure 1C* and *Figure 1D*, black trace) we observed a significant sharpening of the RNAPII occupancy peak compared to the alignment on their primary ACSs (*Figure 1D*, compare red to black traces; *Figure 1C*; *Figure 1—figure supplement 2c* for the statistical significance of the signal loss over the secondary ACSs). This suggests that RNAPII is indeed pausing immediately upstream of the secondary ACS. Interestingly, when we aligned polyadenylated RNA 3'-ends using the first position of the predicted secondary ACSs, we observed that transcription termination distributed preferentially ≈50nt after the anchor (*Figure 1E*, blue trace, compare to RNAPII distribution, black trace; see also *Figure 1—figure supplement 2E*) indicating that in most instances antisense transcription terminates downstream of the site of RNAPII pausing.

To better highlight the presence and the role of a roadblock (RB) at these origins, we examined local transcription by RNAPII CRAC under conditions in which an essential component of either the CPF-CF or the NNS termination pathways is affected, that is in an *rna15-2* mutant at the non-permissive temperature, or by depleting Nrd1 by the auxin-degron method (*Candelli et al., 2018*). We reasoned that defects in CPF-CF or the NNS pathways would affect the levels of neighboring readthrough transcription directed toward these origins and consequently increase the transcriptional loads challenging the roadblocks. Representative examples are shown in *Figure 2*.

In the case of *ARS305* (*Figure 2A*), low levels of readthrough transcription are found at the terminators of the adjacent transcription units (*YCL049C* or CUT040) and are subjected to roadblock termination at both the main (blue) or the putative secondary ACSs (red, overlaps with the previously mapped B4 element (*Huang and Kowalski, 1996*)), respectively. Increase in readthrough transcription at the *YCL049C* gene in *rna15-2* cells (sense transcription, light green track) or at CUT040 upon Nrd1 depletion (antisense transcription, light pink track), leads to increased accumulation of RNAPII at both ACSs and to transcription invading the ARS.

Two ACSs were previously proposed for *ARS413* (*Figure 2B*): sense ACS1 (*Eaton et al., 2010*) and antisense ACS2 (*Nieduszynski et al., 2006*). Transcription on the plus strand is strongly roadblocked at ACS1, while transcription on the minus strand is roadblocked at both ACS2 and ACS1. In both cases, transcription derives only from the upstream genes (*YDL073W* and *YDL072C*, respectively) because no additional initiation sites could be detected, even in cytoplasmic and nuclear RNA degradation mutants (data not shown). When the transcription load was increased by affecting the termination of *YDL073W* and *YDL072C* in *rna15-2* cells at the non-permissive temperature (light green tracks), RNAPII occupancy at the RBs increases and some readthrough within the ARS occurs.

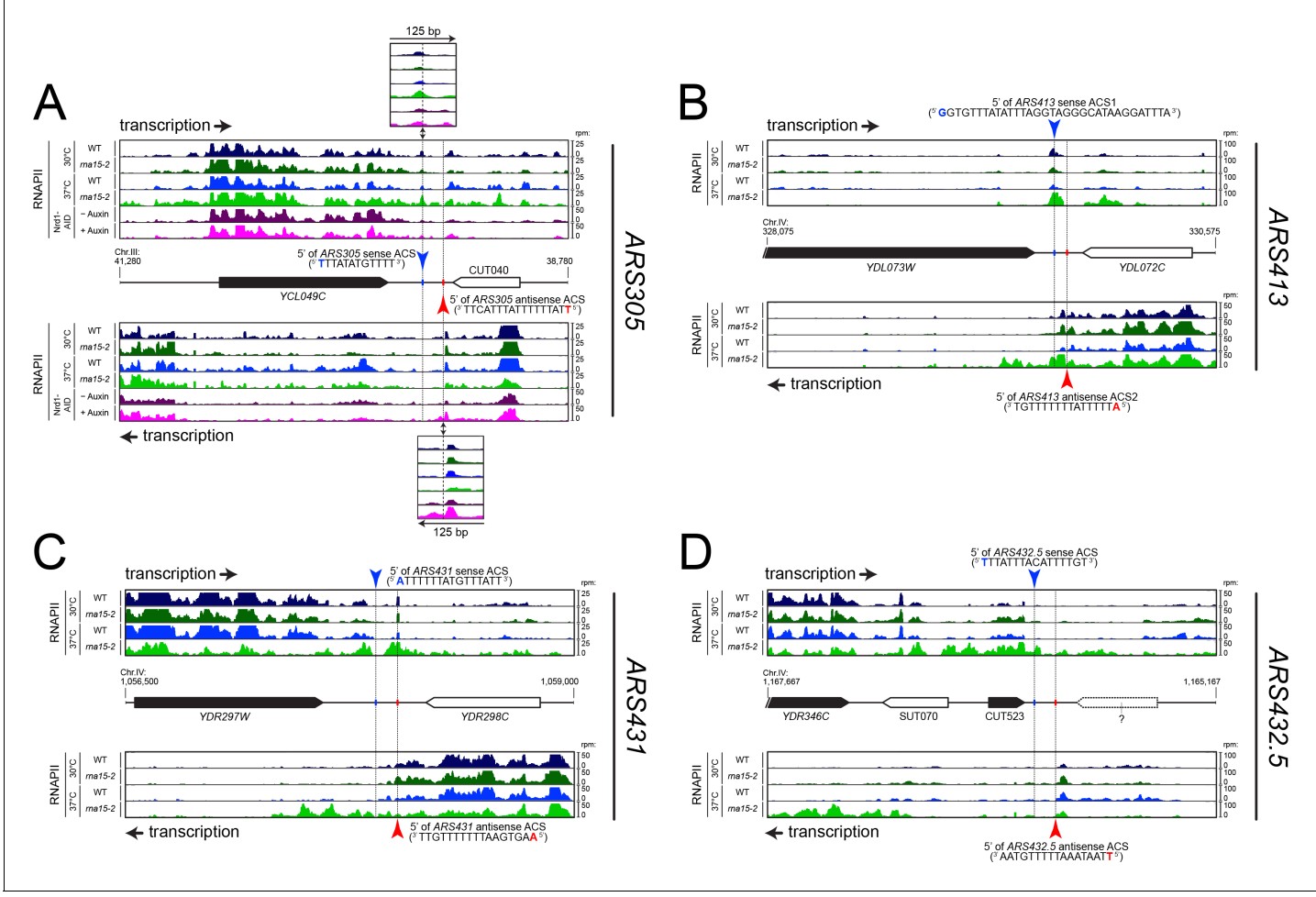

**Figure 2.** RNAPII occupancy at individual ARS detected by CRAC analysis. RNAPII occupancy at sites of roadblock detected upstream *ARS305* (**A**), *ARS413* (**B**), *ARS431* (**C**) and *ARS432.5* (or *ARS453*, (**D**) by CRAC (*Candelli et al., 2018*). The pervasive transcriptional landscape at these ARSs is observed in wild-type cells (WT, blue) or cells bearing a mutant allele for an essential component of the CPF-CF transcription termination pathway (*rna15-2*, green) at permissive (25°C, dark colors) or non-permissive temperature (37°C, light colors). In the case of *ARS305* (**A**), RNAPII occupancy is also shown in cells rapidly depleted for an essential component of the NNS transcription termination pathway through the use of an auxin-inducible degron tag (Nrd1-AID; (−) Auxin: no depletion, dark pink; (+) Auxin: depletion, light pink).

DOI: https://doi.org/10.7554/eLife.40802.005

This example suggests that both ACSs are occupied by the ORC complex, although it is not clear whether they function in conjunction or alternatively in different cells.

Two additional examples are shown in *Figure 2*. In the case or *ARS431* (*Figure 2C*), the RB is more prominent on the site of the primary ACS and increases when the transcriptional load is higher due to readthrough from the upstream gene, *YDR297W*, in *rna15-2* cells. On the contrary, a prominent site of RB at the secondary ACS is observed at *ARS453* (or *ARS432.5*; *Figure 2D*), while the RB at primary ACS cannot be observed because transcription of CUT523 appears to terminate efficiently upstream.

Taken together, these results suggest that primary and secondary ACSs, both presumably bound by ORC, can induce RNAPII pausing at the borders of replication origins. However, while RNAPII generally pauses and terminates upstream of primary ACS sequences, RNAPII often pauses at secondary ACS but terminates downstream. Importantly, such ARS footprint in the pervasive transcription landscape (*Figure 2*) provides independent *in vivo* evidence of the role of secondary ACS sequences (*Coster and Diffley, 2017*), while our meta-analyses (*Figure 1*) strongly suggest a general functional difference between primary and secondary ACSs with regards to incoming transcription.

# Termination of transcription at ARSs is mediated by ORC binding to the DNA

Transcription termination around origins might depend on many termination factors. The main transcription termination pathways in *S. cerevisiae*, NNS- and CPF-dependent, rely on the recognition of termination signals on the nascent RNA. Release of the polymerase occurs therefore after the termination signals that have been transcribed and recognized. Transcription termination by roadblock, on the other hand, ensues from a collision of the transcription elongation complex with a DNA bound protein, and therefore occurs upstream of the termination signal. Another characteristic feature of roadblock termination is that the released RNA is subject to exosome-dependent degradation. Both features, termination upstream of the termination signal and nuclear degradation of the released transcripts, are compatible with the notion that roadblock termination occurs at origins. Still, it remains possible that termination at the immediate borders of origins depends on conserved external signals allowing the recruitment of CPF- or NNS- components. According to the position of RNAPII pausing, the most likely roadblocking factor would be the ORC complex bound to the ACS.

We therefore first verified that termination depends on the ACS sequence and to this end we cloned a 500 bp DNA fragment containing *ARS305* in a reporter system allowing the detection of transcription termination (*Porrua et al., 2012*) (*Figure 3*). This fragment conferred ACS-dependent mitotic maintenance to a centromeric version of the reporter construct, indicating that it is a functional ARS (*Figure 3—figure supplement 1*). In this system, a test terminator sequence is cloned between two promoters, the downstream of which allows the expression of a reporter gene, *CUP1*, which is required for yeast growth in the presence of copper ions (*Figure 3A*). Transcription from the upstream promoter interferes with and thus inactivates the promoter driving expression of *CUP1* unless the test sequence contains a terminator. Copper resistant is therefore a reliable, positive read out of the presence of a transcription terminator in the cloned sequence. Consistent with the notion that termination occurs at replication origins, insertion of *ARS305* in the orientation dictated by the T-rich strand of the ACS conferred robust copper-resistant growth to yeast cells (*Figure 3B*), Importantly, copper resistance was abolished when the ACS was mutated, strongly suggesting that termination is strictly dependent on the integrity of the ORC binding site.

This notion was further supported by Northern blot analysis of the transcripts produced when a shorter *ARS305* fragment containing the ACS and the downstream 154nt were introduced in the same reporter construct (*Figure 3C*). A short transcript witnessing the occurrence of termination was readily detected in the presence of *ARS305* (lane 3). Consistent with the notion that roadblock termination occurs at *ARS305*, the transcript released was subject to exosomal degradation and was stabilized by deletion of Rrp6 (lane 4). This short RNA disappeared when the ACS sequence was mutated, to the profit of a longer species resulting from termination downstream of *ARS305*, confirming the ACS-dependency of termination (lane 5). *ARS305* contains, in addition to the ACS, two motifs, B1 and B4, required for full origin function (*Huang and Kowalski, 1996*). Interestingly, B4 is located roughly 100nt downstream of the ACS, and coincides with a predicted secondary ACS required for efficient symmetrical loading of the pre-RC (*Figure 2* and Table 2) (*Coster and Diffley, 2017*). To assess whether the primary ACS is sufficient to induce transcription termination, we mutated both B1 and B4, alone or in combination, and assessed the level of termination by Northern blot. As shown in lanes 6 and 7, mutation of B4 had the strongest effect on termination, which was very similar to the effect observed when the main ACS was mutated. Mutation of B1 had a minor but significant effect. From these experiments, we conclude that the high-affinity ORC-binding site alone is necessary but not sufficient for inducing transcription termination at *ARS305*, and that the secondary ACS (B4) and the B1 motif are additionally required.

To provide independent evidence that ORC bound to the ARS triggers transcription termination by a roadblock mechanism, we took advantage of the finding that many sequences with a perfect match to the ACS consensus do not bind ORC. We used published coordinates of ACSs bound (ORC-ACSs) or not recognized (nr-ACSs) by the ORC complex in ORC-ChIP-seq experiments (*Eaton et al., 2010*), and mapped transcripts 3'-ends (*Roy et al., 2016*) as a proxy for the occurrence of transcription termination (*Figure 4A and B*). As previously, we oriented each ARS according to the direction of the T-rich ORC-ACS or nr-ACS. As expected, the distribution of transcription termination events around the set of ORC-bound ACSs is very similar to the one observed around replication origins mapped by *Nieduszynski et al. (2006)* (compare *Figure 4A* and *Figure 1B*). As in the

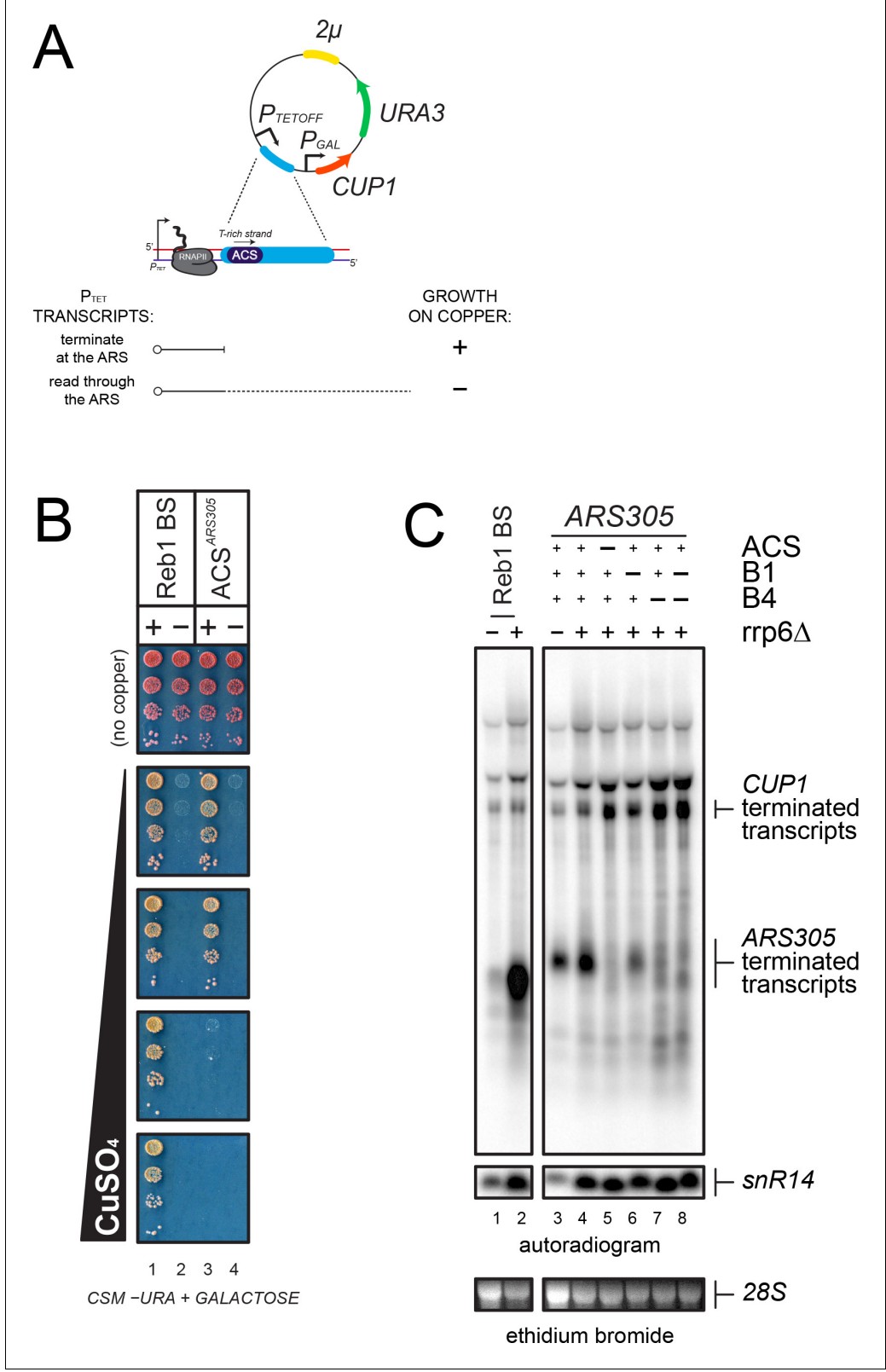

**Figure 3.** Analysis of transcription termination at *ARS305*. (**A**) Scheme of the reporter system (*Porrua et al., 2012*) used to assess termination at *ARS305*. $P_{TETOFF}$: doxycycline-repressible promoter; $P_{GAL}$: *GAL1* promoter. Termination of transcription at a candidate sequence (blue) allows growth on copper containing plates while readthrough transcription inhibits the *GAL1* promoter and leads to copper sensitivity, as indicated. (**B**) Growth

*Figure 3 continued on next page*

*Figure 3 continued*

assay of yeasts bearing reporters containing a Reb1-dependent terminator, (*Colin et al., 2014*, used as a positive control), or *ARS305* (lanes 1 and 3, respectively). Variants containing mutations in the Reb1 binding site (Reb1 BS '−') or the ACS sequence are spotted for comparison (lanes 2 and 4, respectively). (C) Northern blot analysis of $P_{TET}$ transcripts produced in wild-type and *rrp6Δ* cells from reporters containing either a Reb1-binding site (Reb1 BS, lanes 1–2) or wild-type or mutant *ARS305* sequences, as indicated (lanes 3–8). Transcripts terminated within *ARS305* or at the *CUP1* terminator are highlighted.

DOI: https://doi.org/10.7554/eLife.40802.006

The following figure supplement is available for figure 3:

**Figure supplement 1.** ARS305 sequence confers mitotic maintenance to a centromeric plasmid when transcription is shut down.

DOI: https://doi.org/10.7554/eLife.40802.007

previous analysis, many unstable transcripts are produced by termination around origins as witnessed by the overall higher level of 3'-ends mapped in an exosome-deficient strain (*Figure 4A*). The distribution of RNA 3'-ends around the set of nr-ACSs is however radically different, with transcription events presumably crossing the nr-ACS in both directions and terminating downstream (*Figure 4B*). Interestingly, at nr-ACSs, the amounts of 3'-ends detected are very similar in wild-type conditions or upon depletion of both Rrp6 and Dis3 subunits of the nuclear exosome, indicating that termination downstream of nr-ACSs does not produce unstable transcripts and is presumably dependent on the CPF pathway (*Figure 4B*).

Because the ACS sequence is nearly identical in the two datasets, it is unlikely that it alone could be responsible for the termination pattern observed at ORC-ACSs. These observations are consistent with the notion that the presence of ORC bound to the ACS is necessary to roadblock transcribing RNAPII, which releases a fraction of unstable RNAs. To substantiate these findings we set up to assess directly the impact of ORC depletion on transcribing RNAPII at two model origins, *ARS404* and *ARS1004*, located downstream of the *YDL227C* and *YJL217W* genes, respectively. In both cases, RNAPII signals are present immediately upstream of the T-rich strand of the ACS, presumably because of transcription events reading through the upstream terminator that are roadblocked at the site of ORC binding (*Figure 4C*). To assess the efficiency of the roadblock we measured RNA levels immediately upstream and downstream of the T-rich strand of each ACS in a strand-specific manner by RT-quantitative PCR (*Figure 4C and D*). Because no transcription initiation can be detected at either one of the two ACSs (data not shown), RNA signals detected downstream of the ACS are most likely due to molecules that initiate upstream and cross the ACS. We therefore expressed the efficiency of the roadblock as the ratio between the signals downstream and upstream of the ACS. Release of the roadblock is expected to increase this ratio because more RNAPII molecule would traverse the ACS. To affect binding of ORC to the ACS, we used two thermosensitive mutants of two ORC subunits, Orc2-1 and Orc5-1, which affect the binding of ORC to the DNA (*Santocanale and Diffley, 1996*; *Loo et al., 1995*; *Yuan et al., 2017*; *Shimada et al., 2002*). As shown in *Figure 4D*, ORC roadblock at *ARS404* and *ARS1004* is efficient, allowing only between 1–10% of the incoming transcription to cross the ACS in wild-type cells or under permissive temperature for all mutants (*Figure 4D*, 23°C). When the binding of ORC to the ACS was affected in *orc2-1* and *orc5-1* cells at 37°C, a marked increase in the fraction of RNAPII going through the roadblock is observed, indicating that binding of the ORC complex to the ACS is necessary to terminate upstream incoming transcription.

Cdc6 binds DNA cooperatively with ORC and contributes to origin specification by participating to pre-RC assembly (*Speck et al., 2005*; *Speck and Stillman, 2007*; *Yuan et al., 2017*) and references therein). The thermosensitive mutant Cdc6-1 (*Hartwell et al., 1973*) which is affected in pre-RC assembly at the restrictive temperature (*Cocker et al., 1996*), still does not preclude ORC to footprint at candidate ARSs (*Santocanale and Diffley, 1996*). Remarkably, the transcriptional roadblock was markedly reduced in a *cdc6-1* mutant at the non-permissive temperature, to a similar extent as for the *orc2-1* and *orc5-1* mutants. This indicates that the assembly of an ORC•Cdc6 complex, or the full complement of the pre-RC at the candidate ARS, is essential for efficiently roadblocking RNAPII.

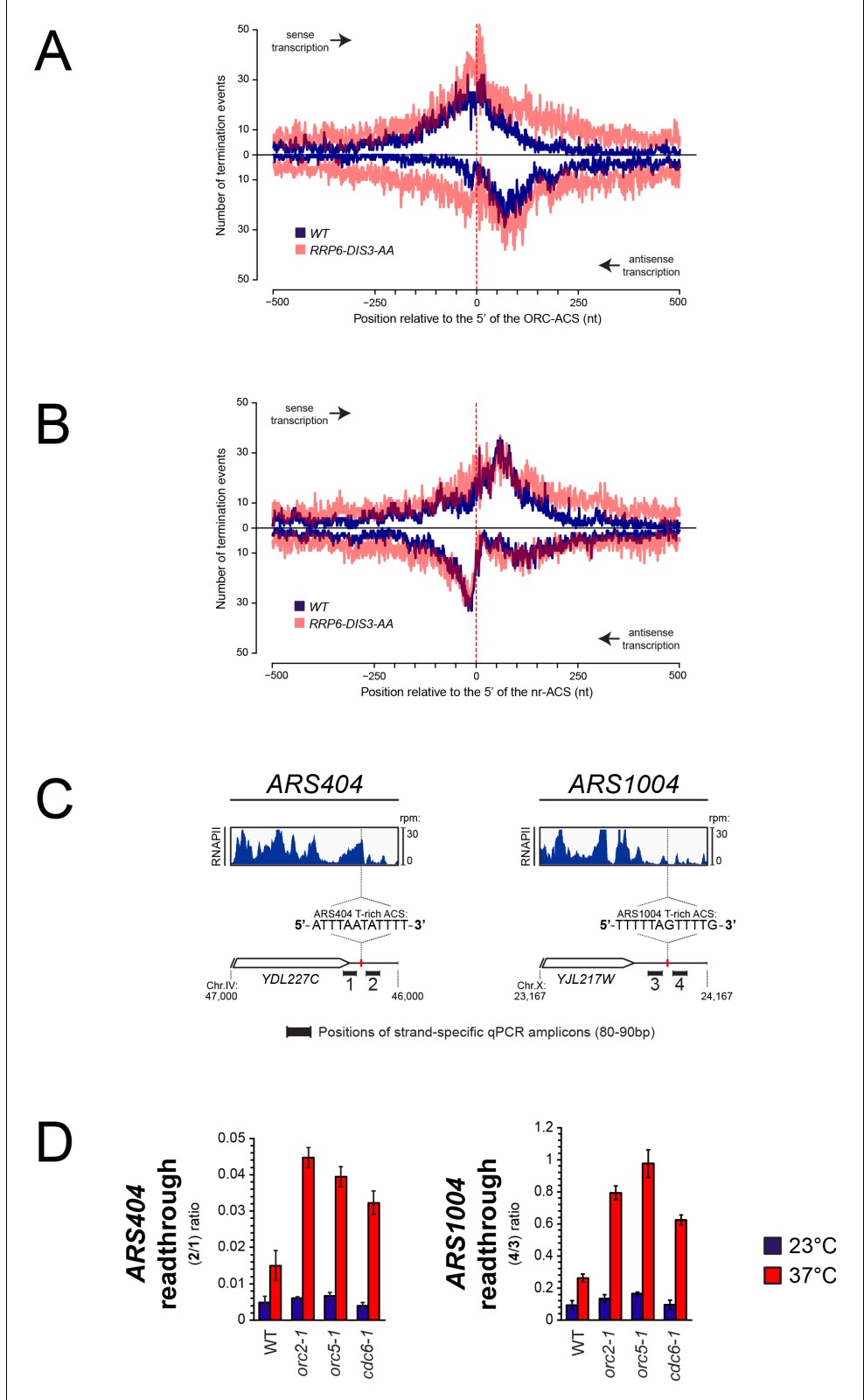

**Figure 4.** Role of ORC in the roadblock of RNAPII at origins. (**A**) Distribution of RNA 3'-ends at genomic regions aligned at ACS sequences recognized by ORC (ORC-ACS) as defined by *Eaton et al. (2010)* (i.e. defined based on the best match to the consensus associated to each ORC-ChIP peak). Each origin was oriented according to the direction of the T-rich strand of its ORC-ACS and regions were aligned at the 5' ends of the ORC-ACSs. As in 1B, RNA 3'-ends (*Roy et al., 2016*) were from transcripts expressed in wild-type cells (blue) or from cells depleted for exosome components

*Figure 4 continued on next page*

*Figure 4 continued*

(transparent red). At each position around the anchor, presence or absence of an RNA 3'-end was scored independently of the read count. Distributions of RNA 3'-ends both on the sense (top) and the antisense (bottom) strands relative to the ORC-ACSs are plotted. (B). Same as in (A) except that genomic regions were aligned at ACS sequences not recognized by ORC (nr-ACS) as defined by *Eaton et al. (2010)* (i.e. defined as ACS motifs for which no ORC ChIP signal could be detected). (C). Quantification of the roadblock at individual ARSs. For each ARS, the snapshot includes the upstream gene representing the incoming transcription. The distribution of RNA polymerase II (dark blue) detected by CRAC (*Candelli et al., 2018*) at *ARS404* (left) and *ARS1004* (right) oriented according to the direction of their T-rich ACS strands is shown. The positions of the qPCR amplicons used for the RT-qPCR analyses in (D) are indicated. (D). RT-qPCR analysis of transcriptional readthrough at *ARS404* and *ARS1004*. Wild-type, *orc2-1*, *orc5-1* and *cdc6-1* cells were cultured at permissive temperature and maintained at permissive (23°C, blue) or non-permissive (37°C, red) temperature for 3 hr. The level of readthrough transcription at *ARS404* (left) or *ARS1004* ACS (right) was estimated by the ratio of RT-qPCR signals after and before the ACS, as indicated. Data were corrected by measuring the efficiency of qPCR for each couple of primers in each reaction. Values represent the average of at least three independent experiments. Error bars represent standard deviation.

DOI: https://doi.org/10.7554/eLife.40802.010

From these results, we conclude that the stable binding of the ORC complex to the ACS is necessary but not sufficient to efficiently terminate incoming transcription at ARS by a roadblock mechanism.

## Impact of local pervasive transcription on ARS function

In spite of the presence of bordering roadblocks, low levels of pervasive transcription, which presumably originates in neighboring regions and cross the sites of ORC occupancy, were detected within replication origins (*Figures 1–3*). To assess the impact of local physiological levels of transcription within ARS, we sought correlations between total RNAPII occupancy on both ARS strands in a window of 100nt starting at the first base of the primary ACS, and licensing efficiency or origin activation (*Hawkins et al., 2013*) We ordered the origins described by *Nieduszynski et al. (2006)* according to the levels of transcription at and immediately downstream of the T-rich ACS and compared the licensing efficiency of the 30 origins having the highest transcription levels to the rest of the population (160 origins) for which replication metrics were available (total of 190 origins) (*Supplementary file 1* Table 1). We found that the efficiency of licensing was significantly lower for the origins having the highest levels of transcription (*Figure 5A*; p = 0.003). We also found that origins having the highest levels of transcription display a lower probability of firing compared to the rest of the population (*Figure 5B*; p = 0.012).

The effect observed on origin firing might be a consequence of the impact of transcription on licensing. However, it is also possible that local levels of pervasive transcription impact origin activation after licensing. To address this possibility, we focused on the 30 origins that have the highest levels of incoming transcription as defined by the levels of RNAPII occupancy preceding (*Figure 6A*; 'A') and following (*Figure 6A*; 'C') a 200nt window aligned at the 5' end of the ACS (*Figure 6A*; 'B') (*Supplementary file 1* Table 2, *Supplementary file 1* Table 3). Consistent with the previous analyses performed on all origins, transcription over 'B' strongly anticorrelated with origin competence (p = $2*10^{-4}$; *Figure 6B*) and efficiency (p = $5*10^{-5}$; *Figure 6C*). When we plotted the probability of licensing ($P_L$) against the probability of firing ($P_F$), we identified two classes of origins: the first that aligns almost perfectly on the diagonal ($R^2$ = 0.99; *Figure 6D*, red) contains origins that fire with high probability once licensed. The second contains on the contrary origins firing with a lower probability, even when efficiently licensed (*Figure 6D*, black). As the probability of firing ($P_F$) is the product of the probability of licensing ($P_L$) by the probability of firing once licensing has occurred ($P_{F|L}$), the latter is defined by the ratio $P_F/P_L$. We then sought correlations between the total level of transcription over each ARS and the efficiency with which it is activated at the post-licensing step ($P_{F|L}$). Strikingly, origins that have a high $P_{F|L}$ are generally insensitive to transcription (*Figure 6E*, red); on the contrary, origins that have a low $P_{F|L}$ are markedly sensitive to the levels of overlapping transcription ($R^2$ = 0.55; p = 0.002; *Figure 6E*, black). This generally holds true when the median time of firing (*Hawkins et al., 2013*) is considered: origins with a high $P_{F|L}$ are generally firing earlier and in a manner that is independent from transcription levels over B (*Figure 6F*, red), while, conversely, origins that have a low $P_{F|L}$ tends to fire later when transcription over B increases ($R^2$ = 0.44; p = 0.009; *Figure 6F*, black).

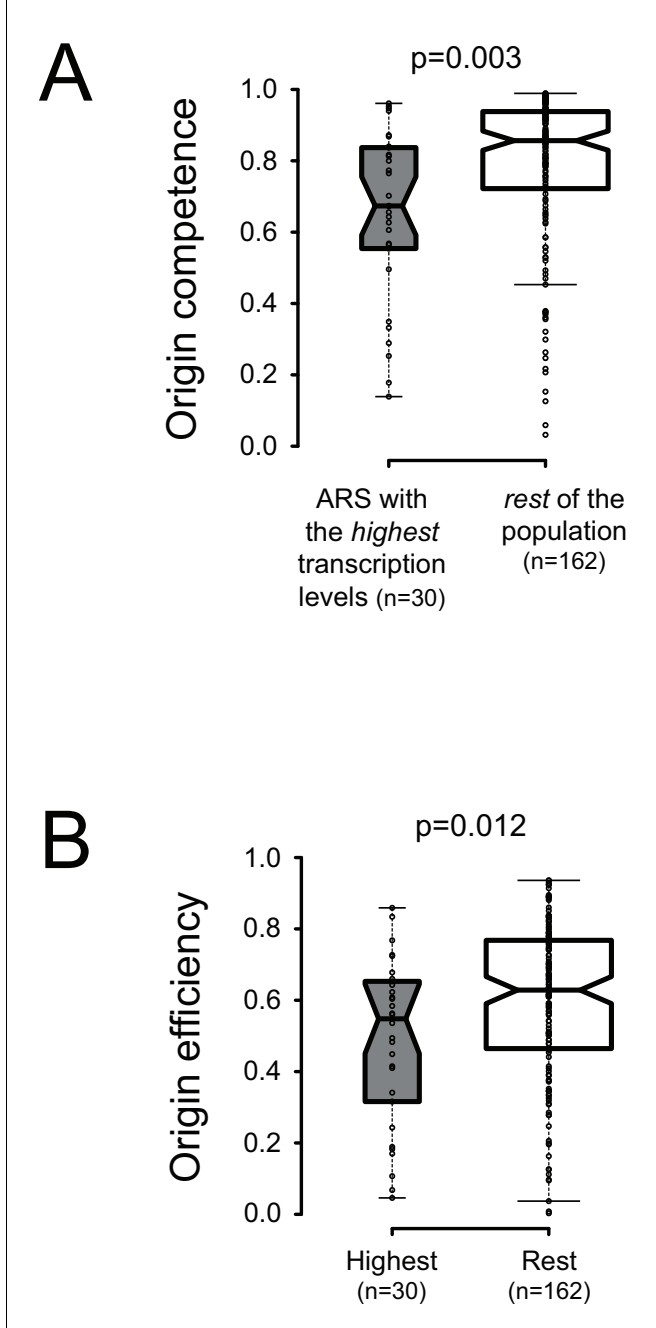

**Figure 5.** Local pervasive transcription impacts origin competence and efficiency. Transcription levels were assessed in the first 100 nt of each ARS, starting at the 5' end of the ACS, by adding RNAPII read counts (*Schaughency et al., 2014*) on both strands of the region. Origins were ranked based on transcription levels and the origins having the highest transcription levels (30/192, grey boxplots) were compared to the rest of the population (162/192, white boxplots). Origin metrics (licensing, 5A, and firing efficiency, 5B) for the two classes of origins were retrieved from *Hawkins et al. (2013)*. Boxplots were generated with BoxPlotR (http://shiny.chemgrid. org/boxplotr/); center lines show the medians; box limits indicate the 25th and 75th percentiles; whiskers extend 1.5 times the interquartile range (IQR) from the 25th and 75th percentiles. Notches are $1.58 \ast IQR/n^{1/2}$.
DOI: https://doi.org/10.7554/eLife.40802.011

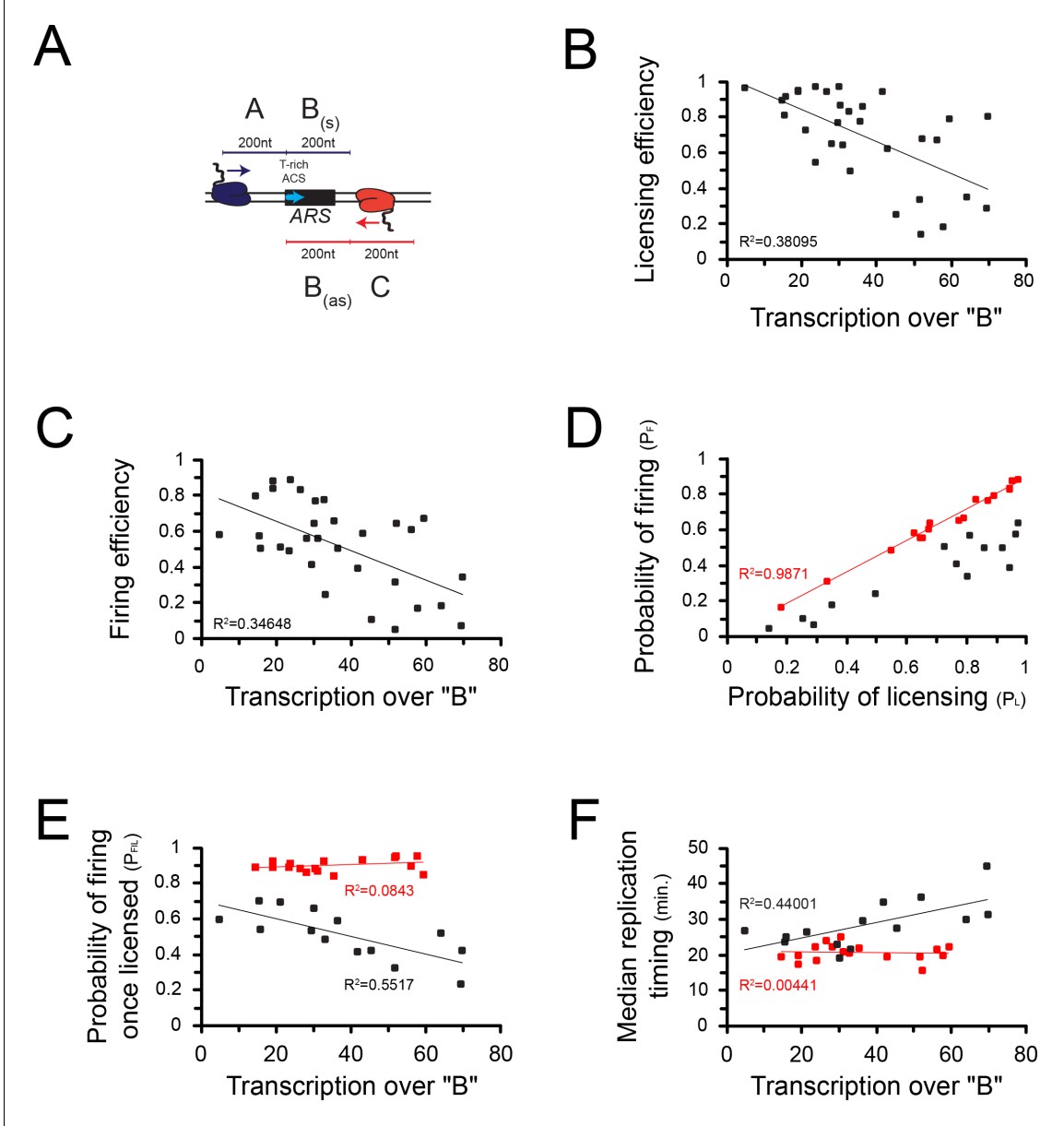

**Figure 6.** Correlations between transcription and origin function. (**A**) Origins were first selected based on the levels of pervasive transcription to which they are exposed, calculated by adding RNAPII reads (*Schaughency et al., 2014*) over the 'A' (sense direction) or the 'C' (antisense direction) regions. For the selected ARSs, levels of pervasive transcription were then calculated over the 'B' region by summing RNAPII reads over the 'B$_a$' (sense direction) and the 'B$_{as}$' (antisense direction) regions, as indicated in the scheme. (**B**) Correlation between transcription over the ARS and origin competence. (**C**) Correlation between transcription over the ARS and origin efficiency. (**D**) Identification of two classes of origins, one that fires with high probability when licensing has occurred (high P$_{F|L}$, red dots) and the other that fires less efficiently once licensed (low P$_{F|L}$, black dots). (**E**) Correlation between P$_{F|L}$ and transcription. The efficiency of firing at the post-licensing step correlates with the levels of pervasive transcription only for origins with low P$_{F|L}$ (black dots). Origins that fire very efficiently once licensing occurred (P$_{F|L} \approx 1$) are generally not sensitive to pervasive transcription (red dots). (**F**) Origins with a low P$_{F|L}$ (black dots) have a firing time that correlates with pervasive transcription, while origins with high P$_{F|L}$ (red dots) fire early independently of pervasive transcription levels.

DOI: https://doi.org/10.7554/eLife.40802.012

We conclude that the efficiency of origin licensing generally negatively correlates with the levels of pervasive transcription within the ARS. Interestingly, a class of origins exists for which the local levels of transcription also impact origin activation after licensing.

## Asymmetry of origin sensitivity to transcription

It has been suggested that the ORC complex binds the secondary ACS with lower affinity relative to the primary ACS (*Coster and Diffley, 2017*). If the affinity of ORC binding to DNA reflected its efficiency at roadblocking RNA polymerases, the existence of both primary and secondary ACSs might imply that incoming transcription upstream of the primary ACS (defined as 'sense' transcription) might be roadblocked more efficiently than incoming transcription upstream of the secondary ACS (defined as 'antisense' transcription). As a consequence, antisense transcription would be more susceptible to affect origin function. To assess the functional impact of this asymmetry, we turned to a natural model case, *ARS1206*, which immediately follows *HSP104*, a gene activated during heat shock (*Figure 7A*).

We cloned the *HSP104* coding sequence and the following *ARS1206* under the control of a doxycyclin-repressible promoter (P$_{TETOFF}$), similar in strength and characteristics to the *HSP104* promoter (*Mouaikel et al., 2013*) (*Figure 7A*). We verified that the HSP104 gene is transcribed and produces a transcript similar in size to the endogenous *HSP104* RNA (data not shown), implicating that transcription termination occurs efficiently in this construct. This is expected to allow origin function, even under conditions of the strong transcription levels induced by the TET promoter. Indeed, after deletion of the ARS present in the plasmid backbone (*ARS1*), the plasmid could still be maintained in yeast cells, showing that it can rely on *ARS1206* for replication (data not shown; *Figure 7D*).

We recently showed that transcription readthrough at canonical terminators is widespread in yeast and is one important component of pervasive transcription (*Candelli et al., 2018*). Although *ARS1206* is active, we predicted that the low levels of transcription reading through the *HSP104* terminator might impact its efficiency in an orientation-dependent manner. To test this hypothesis, we inverted the orientation of *ARS1206* on the plasmid, so that transcription from *HSP104* would approach the origin from its secondary ACS side (*Figure 7A*). We observed equivalent levels of *HSP104* expression from plasmids containing *ARS1206* in the sense (pS) or the antisense (pAS) orientation (*Figure 7B*) and concluded that transcription termination, which would have created unstable RNAs when impaired (*Libri et al., 2002*), occured still efficiently upon *ARS1206* inversion. Consistently, high resolution Northern blot analysis of the 3'-ends of the *HSP104* RNA produced by pS and pAS confirmed that the site of polyadenylation was not altered by inversion of *ARS1206* and no readthrough RNAs could be detected (*Figure 7C*). Strikingly, when pS or pAS were transformed into wild-type cells, and yeasts were grown in a medium non-selective for plasmid maintenance for the same number of generations, *ARS1206* supported plasmid maintenance more efficiently when present on the sense (pS) relative to the antisense (pAS) orientation (*Figure 7D*).

This result is consistent with the notion that constitutive readthrough transcription from the *HSP104* gene affects origin function more markedly when approaching *ARS1206* from the side of the secondary ACS. This result is also consistent with the notion that incoming transcription is roadblocked more efficiently by ORC binding to the primary ACS as opposed to the secondary ACS, in line with the expected lower affinity of the latter interaction. To consolidate this result, we took advantage of previous work demonstrating that the *orc2-1* mutation has a stronger impact on the binding of ORC to ACSs having a poor match to the consensus, even at permissive temperature (*Hoggard et al., 2013*). If binding of ORC to the ACS is the limiting factor for the functional asymmetry we observe, then affecting binding of ORC to the secondary, lower affinity site by the *orc2-1* mutation should exacerbate the instability of the pAS plasmid. Indeed, while pS could be as efficiently maintained in wild-type and *orc2-1* cells, pAS raised only sick uracil auxotroph transformants in the *orc2-1* background, indicating that it could not be efficiently propagated (*Figure 7E*).

We conclude that, while presence of primary and secondary ACSs at origin borders participates to the shielding of origins from pervasive transcription, this protection occurs asymmetrically.

## Discussion

Transcription by RNA polymerase II occurs largely beyond annotated regions and produces a wealth of non-coding RNAs. Such non-coding transcription events have the potential to alter the chromatin landscape and affect in many ways the dynamics of other chromatin-associated processes. They originate from non-canonical transcription start site usage or from transcription termination leakage, as recently shown in the yeast and mammalian systems (*Vilborg et al., 2015*; *Grosso et al., 2015*; *Rutkowski et al., 2015*; *Candelli et al., 2018*). Although the frequency of these events is generally

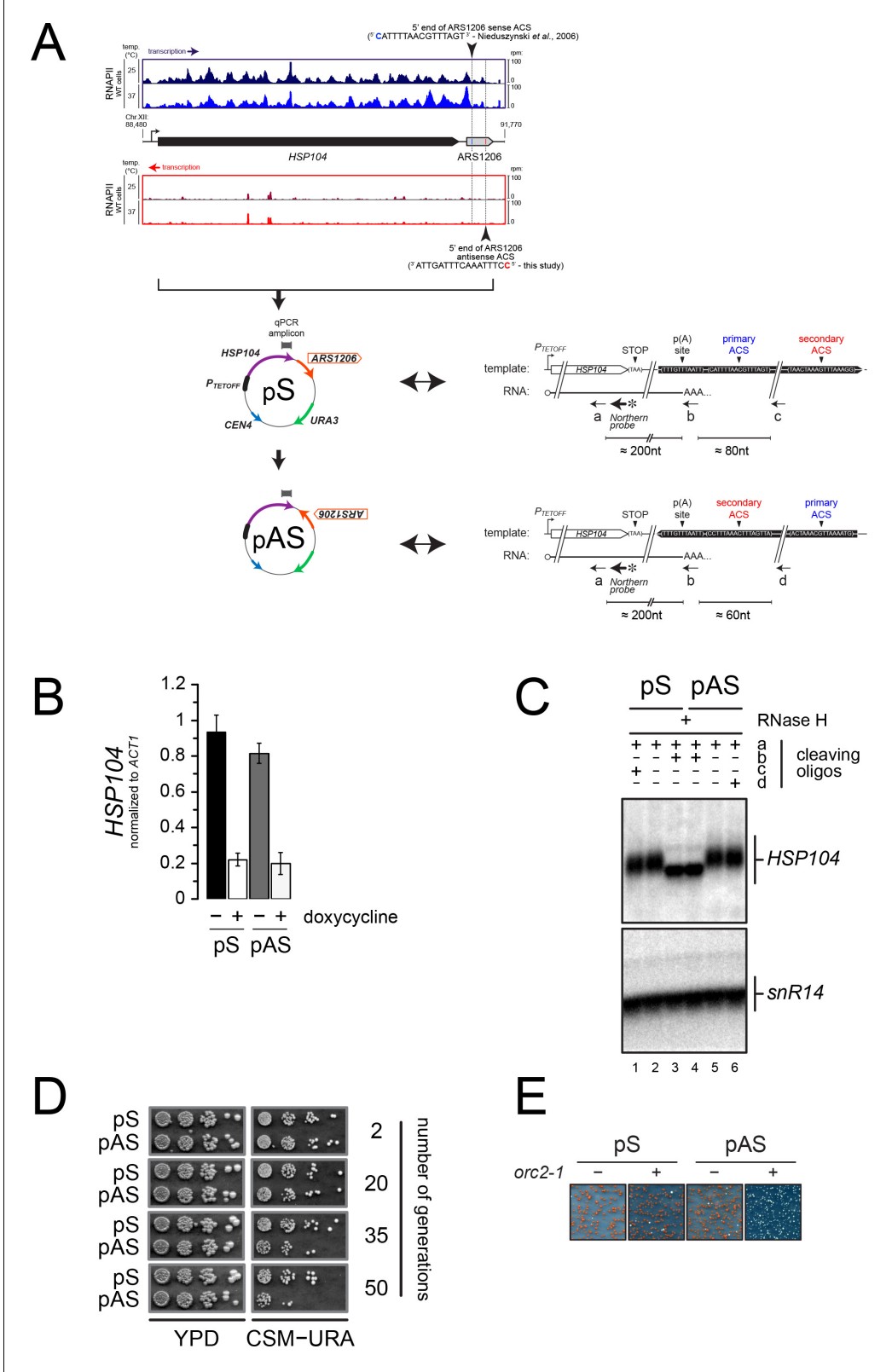

**Figure 7.** Asymmetry of origin sensitivity to pervasive transcription. (**A**) Top: pervasive transcriptional landscape detected by RNAPII CRAC (**Candelli et al., 2018**) at *YLL026W* (*HSP104*) and *ARS1206* in wild-type cells, both on Watson (blue) and Crick (red) strands, at 25°C (dark colors) and 37°C (light colors). The 5' ends and the sequences of the proposed primary ACS and the predicted secondary ACS for *ARS1206* are shown. Bottom: schemes of the reporters containing the *HSP104* gene and *ARS1206* placed under the control of a doxycycline-repressible promoter (*P_TETOFF*). The *Figure 7 continued on next page*

*Figure 7 continued*

position of the amplicon used for the qPCR in (B) is shown. pS and pAS differ for the orientation of *ARS1206*, with the primary (pS) or the secondary ACS (pAS) exposed to constitutive readthrough transcription from *HSP104*. The sequence and the organization of the relevant region are indicated on the right for each plasmid. The positions of the oligonucleotides used for RNaseH cleavage (black arrows) and of the probe used in (C) are also indicated. The sequences of the oligonucleotides is reported in Table 1, with the following correspondence: cleaving oligo 'a'=DL163; Northern probe = DL164; cleaving oligo 'b' = DL473; cleaving oligo 'c' = DL3991; cleaving oligo 'd' = DL3994. (B). Quantification by RT-qPCR of the *HSP104* mRNA levels expressed from pS or pAS in the presence or absence of 5 µg/mL doxycycline. The position of the qPCR amplicon is reported in (A). (C). Northern blot analysis of *HSP104* transcripts extracted from wild-type cells and subjected to RNAse H treatment before electrophoresis using oligonucleotides 'a-d' (positions shown in A). All RNAs were cleaved with oligonucleotide 'a' to decrease the size of the fragments analyzed and detect small differences in size. Cleavage with oligonucleotide 'b' (oligo-dT) (lanes 3, 4) allowed erasing length heterogeneity due to poly(A) tails. Oligonucleotides 'c' and 'd' were added in reactions run in lanes 1 and 6, respectively, to detect possible longer products that might originate from significant levels of transcription readthrough from *HSP104*, if the inversion of *ARS1206* were to alter the transcription termination efficiency. Products of RNAse H degradation were run on a denaturing agarose gel and analyzed by Northern blot using a radiolabeled *HSP104* probe (position shown in A). (D). Stability of plasmids depending on *ARS1206* for replication as a function of ARS orientation. pS or pAS was transformed in wild-type cells and single transformants were grown and maintained in logarythmic phase in YPD for several generations. To assess the loss of the transformed plasmid, cells were retrieved at the indicated number of generations and serial dilutions spotted on YPD (left) or minimal media lacking uracile (right) for 2 or 3 days, respectively, at 30°C. (E). Mutation of *ORC2* affects more severely the stability of pAS compared to pS. Transformation of pS and pAS in wild-type (*ORC2*, '−') or mutant (*orc2-1*, '+') cells. Pictures were taken after 5 days of incubation at permissive temperature (23°C).
DOI: https://doi.org/10.7554/eLife.40802.013

low, the persistence of RNA polymerases is dependent on the speed of elongation and the occurrence of pausing and termination, potentially leading to significant occupancy at specific genomic locations where they could have a function. The crosstalks between transcription and replication have been traditionally analyzed in the context of strong levels of transcription, which, aside from a few specific cases, do not represent the natural exclusion of replication origins from regions of robust and generally constitutive transcription (*MacAlpine and Bell, 2005*; *Nieduszynski et al., 2005*; *Donato et al., 2006*). We studied here the impact of pervasive transcription on the specification and the function of replication origins. We demonstrate that origins have asymmetric properties in terms of the resistance to incoming transcription. The inherent protection of replication origins by transcription roadblocks limits the extent of transcription events within these regions. Nevertheless, polymerases that cross the roadblock borders impact both the efficiency of licensing and origin firing, demonstrating that physiological levels of pervasive transcription can shape the replication program of the cell. Importantly, since the global transcriptional landscape is sensitive to changes dictated by different physiological or stress conditions, pervasive transcription is susceptible to regulate the replication program according to cellular needs.

## Replication initiates in regions of active transcription

Based on the presence and relative orientation of stable annotated transcripts, early studies have concluded that replication origins are excluded from regions of active transcription (*Donato et al., 2006*; *Nieduszynski et al., 2005*). To the light of our results it is clear that this notion needs to be revisited: if origins are generally excluded from regions of *genic* transcription, they dwell in a transcriptionally active environment populated by RNA polymerases that generate pervasive transcription events. These events have multiple origins and are generally of lower intensity relative to *bona fide* genic transcription. When ARSs are located in between divergent genes or more generally upstream of a gene, they might be exposed to natural levels of divergent transcription due to the intrinsic bidirectionality of promoters. When they are located downstream of a gene, they are potentially exposed to transcription naturally reading through termination signals (*Candelli et al., 2018*), which, depending on the level of expression of the gene and the robustness of termination signals, can be consequential.

## Transcription termination occurs around and within origins

Nonetheless, origins are not porous to surrounding transcription and the presence of one ARS generates a characteristic footprint in the local RNAPII occupancy signal. When origins are oriented according to the main ORC binding site, the ACS, RNAPII signal is found to accumulate to some extent, depending on the levels of incoming transcription (*Figures 1A* and *2*), and sharply decrease

in correspondence of the ACS. We provide several lines of evidence supporting the notion that RNAPII is paused at the site of ORC binding and that transcription termination occurs by a roadblock mechanism. First, we observed a relative enrichment of RNA 3'-ends coinciding with the descending RNAPII signal, indicating that termination occurs at or before transcription has proceeded through the termination signal (the ACS). Second, a fraction of the RNAs produced are sensitive to exosomal degradation (*Colin et al., 2014*; *Candelli et al., 2018*). Third, mutation of the ORC-binding site prevents efficient termination in our reporter system. Finally, mutational inactivation of ORC and Cdc6 erases the roadblock and allows transcription to cross the ACS at two natural model origins.

These findings are seemingly in contrast with earlier reports showing that inserting model ARSs in a context of strong transcription leads to transcription termination *within* ARSs independently of the ORC-binding site or other sequence signals required for origin function in replication (*Chen et al., 1996*; *Magrath et al., 1998*). One possibility is that the cloned fragments in these early studies accidentally contain transcription termination signals, some of which were not annotated when these experiments were performed. This is likely the case for ARS305 and ARS209 that both contain a CUT directed antisense to the T-rich strand-oriented ACS. ARS416 (ARS1) and ARS209, also used in these studies, might also contain termination signals from the contiguous *TRP1* and *HHF1* genes, respectively. Another possibility is that transcription termination occurred both at the roadblock site (the ACS) and internally, but the former was missed because of the poor stability of the RNA produced. As discussed below, we also found evidence of internal termination, but preferentially when examining the fate of antisense transcription (i.e. entering the ARS from the opposite side of the main ACS oriented by its T-rich strand).

The transcriptional footprint observed for antisense transcription shows a large peak when origins are aligned on the main ACS but condenses into a well-defined peak when the alignment is done on the presumed secondary ORC-binding sites (*Coster and Diffley, 2017*) (*Figure 1D*), suggesting that RNAPII indeed pauses at these sites. However, transcription termination, inferred from the distribution of RNA 3'-ends, occurs downstream of the putative secondary ACS, within the ARS body (*Figure 1E*). Because these RNAs are stable, we suggest that they are generated by CPF-dependent termination, possibly because RNAPII encounters cryptic termination signals, or because the ARS chromatin environment prompts termination. Whether the occurrence of internal termination has functional implications for origin function is unclear; nevertheless, our analyses suggest that the presence of antisense RNAPIIs within the origin is important for modulating its function (see below).

## Topological organization of replication origin factors detected by transcriptional footprinting

We propose that the asymmetrical distribution of RNAPII at ARS borders relates to the 'quasi-symmetrical' model for pre-RC assembly on chromatin, as proposed by Coster and Diffley (*Coster and Diffley, 2017*). Earlier data suggested that binding of a single ORC molecule at a primary ACS is necessary and sufficient to drive the deposition of one Mcm2-7 double-hexamer (DH) around one DNA molecule (*Ticau et al., 2015*). However, given the topology of ORC binding to DNA (*Lee and Bell, 1997*; *Li et al., 2018*) and the mode of Mcm2-7 deposition around DNA (*Frigola et al., 2013*), a drastic conformational change would be required to assemble one Mcm2-7 DH with only one ORC (*Zhai et al., 2017*; *Bleichert et al., 2018*). The quasi-symmetrical model, in contrast, postulates that two distinct ORC molecules bind cooperatively each ARS at two distinct ACS sequences. One ORC binds the 'primary' ACS to load one half of the pre-RC, while the second ORC binds a 'secondary', degenerate ACS, to load the other half of the pre-RC in opposite orientation (*Yardimci and Walter, 2014*; *Coster and Diffley, 2017*). Each Mcm2-7 hexamer translocating towards the other would then form the Mcm2-7 DH.

The transcriptional footprinting profile around origins shows an antisense RNAPII signal peaking at aligned potential secondary ACSs identified by their match to the consensus (*Coster and Diffley, 2017*), which testifies to the general functional significance of secondary ACSs prediction. The distribution of distances between the two 5' ends of the two ACSs has a mode of 110nt, which is consistent with the expected physical occupancy of at least one Mcm2-7 DH (*Remus et al., 2009*). This distance is also consistent with the optimal distance between the two ACSs for a functional cooperation in pre-RC complex formation *in vitro* (*Coster and Diffley, 2017*). We show that, presumably because of the average lower affinity of ORC binding to the secondary ACS, transcription termination does not occur upstream of the latter but within the ARS, where RNAPII could favor the

translocation of one Mcm2-7 hexamer towards the other, or 'push' a pre-RC intermediate (*Warner et al., 2017*) or the DH away or against the high affinity ORC binding site. On a case-by-case basis, it can be envisioned that antisense transcription might participate to the specification of the position of licensing factors (*Belsky et al., 2015*).

### Functional implications for pervasive transcription at ARS

As highlighted above, early studies examined the impact of transcription on origin function by driving strong transcription through candidate ARSs (*Murray and Cesareni, 1986*; *Snyder et al., 1988*; *Chen et al., 1996*; *Kipling and Kearsey, 1989*), or estimated the transcriptional output at ARSs based on the relative orientation of stable annotated transcripts (*Nieduszynski et al., 2005*; *Donato et al., 2006*). To the light of the recent, more extensive appreciation of the transcriptional landscape, these studies did not address the impact of local, physiological levels of transcription on origin function. Our results demonstrate that the predominant presence of replication origins at the 3'-ends of annotated genes or upstream of promoters in the *S. cerevisiae* genome (*MacAlpine and Bell, 2005*; *Nieduszynski et al., 2005*; *Donato et al., 2006*) does not preclude ARS from being challenged by transcription. Rather, pervasive transcription is likely to play an important role in fine-tuning origin function and influence their efficiency and the timing of activation. Similar conclusions have been recently reported in an independent study by *Soudet et al. (2018)*.

The licensing of origin is predominantly sensitive to transcription within the ARS, which might have been expected. The presence of transcribing polymerases might prevent pre-RC assembly or ORC binding to the ACS (*Mori and Shirahige, 2007*; *Lõoke et al., 2010*). Transcription through promoters has been shown to inhibit de novo transcription initiation by increasing nucleosome occupancy in these regions and lead to the establishment of chromatin marks characteristic of elongating transcription. We propose that transcription though origins might induce similar changes that are susceptible to outcompete binding of ORC and/or pre-RC formation.

Once licensing has occurred, firing ensues a series of steps leading to Mcm2-7 DH activation. It was surprising to observe that firing once licensing has occurred is also sensitive to the levels of local pervasive transcription, possibly implying that post-licensing activation steps are also somehow sensitive to the presence of transcribing RNAPII. An alternative, interesting possibility is that transcription complexes might push the Mcm2-7 DH away from the main site of initiation (*Gros et al., 2015*). As a consequence, the actual position of replication initiation would be altered with a given frequency: replication might still initiate but in a more dispersed manner around the origin and would not be taken into consideration in the computation of initiation events. A final possibility is that pre-RC formation is to some extent reversible, and transcription might alter the equilibrium by occupying ARS sequences at a post-licensing but pre-activation step. The subset of origins that we found to be insensitive to transcription might be less prone to sliding or have a slower rate of pre-RC disassembly, which would make them less likely to be influenced by transcription.

The topological organization of replication origins and transcription units has been studied in many organisms, with the general consensus that the replication program is relatively flexible and adapts to the changing transcriptional environment during development or cellular differentiation in multicellular organisms (*Powell et al., 2015*; *Petryk et al., 2016*; *Pourkarimi et al., 2016*). The rapidly dividing *S.cerevisiae* has maintained some of this adaptation of replication to the needs of transcription, for example during meiotic differentiation (*Blitzblau et al., 2012*). Origin specification, nonetheless, relies on a relatively strict requirement for defined ARS sequences, which is possibly more efficient, but also less flexible for adapting to alterations in the transcription program and more sensitive to pervasive transcription. Transcription termination and RNAPII pausing at origin borders are some of the strategies that shape the local pervasive transcription landscape to the profit of origin function, and mute disruptive interferences into fine tuning of origin efficiency and activity.

## Materials and methods

### Yeast strains - oligonucleotides - plasmids

Yeast strains, oligonucleotides and plasmids used in this study are reported in Table 1.

## Metagene analyses

### RNAPII occupancy

For each feature included in the analysis, we extracted the polymerase occupancy values at every position around the feature and plotted the median over all the values for that position in the final aggregate plot.

### Transcription termination around origins

To estimate the extent of transcription termination around replication origins, we considered the detection of 3'-ends of polyadenylated transcripts as a proxy for termination events. We counted, for each position, the number of origins for which at least one 3'-end could be mapped at that position. We then plotted the final score per-position in the aggregate plot. This allowed considering the occurrence of at least one termination event at a given position while minimizing the impact of the steady state level of the transcripts produced by termination. To assess the statistical significance of the peak observed upstream of the primary ACS, we adopted the H0 hypothesis that termination occurs with the same frequency in the whole region of alignment around the origin. We estimated the expected value based on the frequency of termination events (i.e. presence of at least one 3'-end) in a 100nt window located at position −500 from the primary ACS across all available sites. Using this estimate, we calculated the probability of detecting the number of termination events actually observed at every position using the binomial distribution and correcting for the multiple testing factor (*Benjamini and Hochberg, 1995*).

### Analysis of termination at ORC-ACS and nr-ACS

ORC-ACSs are defined as the best match to the consensus under ORC ChIP peaks (*Eaton et al., 2010*). nr-ACSs are defined as sequences containing a nearly identical motif that are not occupied by ORC as defined by ChIP analysis (*Eaton et al., 2010*).

## Correlation between transcription and replication metrics

For the boxplot analyses shown in *Figure 5*, we selected 190 origins out of the 228 described in *Nieduszynski et al. (2006)* for which replication metrics were available (*Hawkins et al., 2013*) and considered the RNAPII read counts in the 100nt following the 5' end of the ACS, in the sense and antisense direction (*Supplementary file 1* Table 1). Origins were ranked based on the transcription levels to establish two groups, one of high and one of low transcription, which were compared in terms of licensing and firing efficiencies. A Student t-test (two tailed, same variance, unpaired samples) was used to estimate the statistical significance of the differences between the two distributions of values.

For the correlation analyses shown in *Figure 6*, we selected origins with the highest levels of incoming transcription by considering a total coverage higher than 10 read counts in an area of 200 bp upstream of the area of origin activity, both on the T-rich and A-rich strand of the ACS consensus sequence (regions 'A' and 'C', *Figure 5*) (*Supplementary file 1* Table 2). Then we summed the total read coverage over the area of origin activity (region 'B', *Figure 5*) on both sense and antisense strand (*Supplementary file 1* Table 3). This value was then correlated with different measures of replication activity.

## Secondary ACS mapping

The coordinates of the predicted secondary ACSs are reported in Table 2. To map putative secondary ACS sequences, we considered a nucleotide frequency matrix for the ACS consensus sequence (*Coster and Diffley, 2017*) and produced a PWM (*Position Weight Matrix*) using the function PWM from the R Bioconductor package 'biostrings' using default options. We used the 'matchPWM' function from 'biostrings' to look for the best match for putative secondary ACSs in the range between the position +10 to+400 relative to the main ACS. We then calculated the distribution of distances between the main and the putative secondary ACSs and the distribution of matching scores (*Figure 1—figure supplement 1*). For the meta-analyses shown in *Figure 1D–E*, we restricted this analysis to a shorter range, considering that secondary ACSs located less than 70nt or more than 200nt might not be biologically significant. The position and scores of all putative sense and antisense ACSs used for the metaanalyses are shown in Table 2.

## Plasmid constructions

Oligonucleotides used for cloning and plasmids raised are reported in Table 1. P$_{TETOFF}$-*HSP104::ARS305::HSP104* P$_{GAL1}$-*CUP1* (*2μ*, *URA3*) plasmids were constructed by inserting a 548 bp fragment containing the wild-type *ARS305*, as defined in OriDB v2.1.0 (http://cerevisiae.oridb.org; chrIII:39,158–39,706) in vector pDL454 (*Porrua et al., 2012*) by homologous recombination in yeast cells. *ARS305* was PCR amplified from genomic DNA using primers DL3370 and DL3371 (*Figure 3B*) or DL3581 and DL3583 (*Figure 3C*). Mutations in *ARS305* were obtained by inserting linkers by stitching PCR and homologous recombination in yeast in regions A, B1 and B4 corresponding to Lin4, Lin22 and Lin102, respectively (*Huang and Kowalski, 1996*).

P$_{TETOFF}$-*HSP104-ARS1206* (pDL214) plasmid was constructed by inserting the *HSP104* gene and the downstream genomic region containing the *HSP104* terminator and *ARS1206* into pCM188 (*ARS1*, *CEN4*, *URA3*) by homologous recombination in yeast. ARS1 was removed from pDL214 by cleavage with NheI and repaired by homologous recombination using a fragment lacking ARS1 to obtain 'pS'. P$_{TETOFF}$-*HSP104-6021sra* (or 'pAS') was constructed by reversing *ARS1206* orientation in 'pS' using homologous recombination in yeast.

## RNA analyses

RNAs were prepared by the hot phenol method as previously described (*Libri et al., 2002*). Northern blot analyses were performed with current protocols and membranes were hybridized to the indicated radiolabeled probe (5'-end labelled oligonucleotide probes or PCR fragments labeled by random-priming in ULTRAhyb-Oligo or ULTRAhyb ultrasensitive hybridization buffers (Ambion)) at 42°C overnight. Oligonucleotides used for generating labeled probes are reported in Table 1. RNase H cleavage was performed by annealing 50pmoles of each oligonucleotide to 20 μg of total RNAs in 1X RNase H buffer (NEB) followed by addition of 2U of RNase H (NEB) and incubation at 30°C for 45 min. Reaction was stopped by addition of 200 mM sodium-acetate pH 5.5 and cleavage products were phenol extracted and ethanol precipitated. Pellets were resuspended in one volume of Northern sample loading buffer and the equivalent of 10 μg of total RNAs were analyzed by Northern blot on a 2% TBE1X agarose gel. Oligonucleotides used for RNase H cleavage assay are reported in Table 1.

For RT-qPCR analyses, RNAs were reverse transcribed with 200U of M-MLV reverse transcriptase (ThermoFisher) and strand specific primers for 45 min at 37°C. Reactions were diluted 10 times before qPCR analyses. Quantitative PCRs were performed on a LightCycler 480 (Roche) in 384-Multiwell plates (Roche) in 10 μL reactions that contained 1% of the reverse transcription mix and 0.25 pmoles of each priming oligonucleotides. Quantification was performed using the ΔΔCt method. 'No RT' controls were systematically analyzed in parallel. Each transcription level reported represents the mean of three independent RNA extractions each assayed in duplicate qPCRs. Error bars represent standard deviations. Oligonucleotides used for RT-qPCR are reported in Table 1. Unless indicated otherwise, transcription levels were normalized to *ACT1* mRNA levels.

## Plasmid-loss assay

Cells were transformed with the indicated *ARS1206*-borne (*CEN4*, *URA3*) plasmid and plated on complete synthetic medium lacking uracile. Single transformants were used to inoculate liquid cultures of CSM −URA that were grown to saturation. Saturated cultures were back diluted into rich medium and maintained in logarythmic phase (i.e. below 0.8 OD$_{600}$) for the indicated number of generations. Aliquots were pelleted, rinsed with water and seven-fold serial dilutions were spotted on YPD and CSM −URA, starting at 0.3 OD$_{600}$. Growth on YPD plates was used to infer that the same numbers of cells were spotted, while reduced numbers of cells growing on CSM−URA reflected plasmid loss over the indicated number of generations.

## Datasets

Datasets used in this study are available from GEO with accession numbers GSE56435 (*Schaughency et al., 2014*), GSE75586 (*Roy et al., 2016*) and GSE97913 (*Candelli et al., 2018*).

## Tables

*Table 1* and *Table 2*.

**Table 1.** Yeast strains, oligonucleotides and plasmids used in this work.

| Yeast strains | Name | Genotype | | Origin |
|---|---|---|---|---|
| | DLY671 | *W303-1a trp1Δ* | | Libri laboratory (BMA64) |
| | DLY2923 | *W303-1a ORC2 ORC5 CDC6* | | Gift from the Pasero laboratory (PP2583) |
| | DLY2685 | *As W303-1a, ORC2 ORC5 cdc6-1* | | Gift from the Schwob laboratory (E589) |
| | DLY2687 | *As W303-1a, orc2-1 ORC5 CDC6* | | Gift from the Schwob laboratory (E1507) |
| | DLY2688 | *As W303-1a, ORC2 orc5-1 CDC6* | | Gift from the Schwob laboratory (E4649) |
| **Oligonucleotides** | **Name** | **Sequence** | | **Purpose** |
| | DL3370 | CATCCACAATTACAACCT ATACATATTCTAGCTGCCTTCA TTGAAACGGCGACGCCC GACGCCGTAATAAC | | Amplification of ARS 305 from genomic DNA. Fw primer bearing 48 bp of homology with DL1702. |
| | DL3371 | gaatctttcttcgaaatc acctttgtatttagcacctgcggtt aatgcggATATATCAGAAACAT ACATATG | | Amplification of ARS305 from genomic DNA. Rev primer bearing 50 bp of homology with DL1666. |
| | DL3446 | CATCCACAATTACAACCT ATACATATTCTAGCTGCCTTCA TTGAAACGATATATCAGAAA CATACATATG | | Insertion of ARS305 in reverse orientation (compare with primer pair DL3370/DL3371). Rev primer bearing homology with DL1702. |
| | DL3447 | gaatctttcttcgaaatcaccttt gtatttagcacctgcggttaatgcggGCG ACGCCCGACGCCGTAATAAC | | Insertion of ARS305 in reverse orientation (compare with primer pair DL3370/DL3371). Fwd primer bearing homology with DL1666. |
| | DL3581 | gaatctttcttcgaaatcacct ttgtatttagcacctgcggttaatgcggGTTTCA TGTACTGTCCGGTGTGATT | | Insertion of shortened ARS305, fwd (cf. DL3447). Primes 32 bp downstream B4 element, removing 291 bp of ARS305 "full-length "3' end. |
| | DL3583 | CATCCACAATTACAAC CTATACATATTCTAGC TGCCTTCATTGAAAC GGAGTATTTGATCCTTTTTTTTATTGTG | | Insertion of shortened ARS305, rev (cf. DL3446). Primes 34 bp upstream ARS305 ACS, removing 83 bp of ARS305 "full-length "5' end. |
| | DL3376 | TTATTCCTCGAGGAC TTTGTAGTTCTTAAAGC | | Insertion of linker substitution Lin102 (B4-) in ARS305 by two stages overlapping PCRs. Fw primer, pair with DL3371. |
| | DL3377 | CTACAAAGTCCTCGA GGAATAATAAATCACACCGGAC | | Insertion of linker substitution Lin102 (B4-) in ARS305 by two stages overlapping PCRs. Rev primer, pair with DL3370. |
| | DL3378 | GGGACCTCGAGGAATA CATAACAAAACATATAAAAACC | | Insertion of linker substitution Lin22 (B1-) in ARS305 by two stages overlapping PCRs. Fw primer, pair with DL3371. |
| | DL3379 | GTTATGTATTCCTCGAG GTCCCTTTAATTTTAGGATATG | | Insertion of linker substitution Lin22 (B1-) in ARS305 by two stages overlapping PCRs. Rev primer, pair with DL3370. |
| | DL3380 | CATAACCCTCGAGG TAAAAACCAACACAATAAAAAAAAGG | | Insertion of linker substitution Lin4 (A-) in ARS305 by two stages overlapping PCRs. Fw primer, pair with DL3371. |
| | DL3381 | GGTTTTTACCTCGAG GGTTATGTATTGTTTATTTTCC | | Insertion of linker substitution Lin4 (A-) in ARS305 by two stages overlapping PCRs. Rev primer, pair with DL3370. |
| | DL1359 | CCTTATACATTAGGTCCTTT | | *HSP104* Northern PCR probe, fwd. Primes about 100nt upstream *HSP104* ATG in PTE TOFF-*HSP104* plasmid serie |
| | DL1360 | ATCCCCCGAATTGATCCGG | | *HSP104* Northern PCR probe, rev. Primes upstream BamHI site in PTETOFF-*HSP104* plasmid serie |
| | DL377 | ATGTTCCCAGGTATTGCCGA | | *ACT1* Northern PCR probe/RT qPCR amplicon, fwd. |

*Table 1 continued on next page*

| Oligonucleotides | DL378 | acacttgtggtgaacgatag | *ACT1* Northern PCR probe/RT qPCR amplicon, rev. |
|---|---|---|---|
| | DL2627 | ATTCAAAAGCGAACACCGA ATTGACCATGAGG AGACGGTCTGGTTTAT | *snR14* Northern oligo probe |
| | DL3763 | CTGGTTGAAACA AATCAGTGCCGGTAAC | ARS404 qRT-PCR, amplicon downstream ARS404 ACS. 5' primes 202 bp after SSB1 STOP, pair with DL3764. |
| | DL3764 | GACTTTTTCTTAACTA GAATGCTGGAGTAGAAATACGC | ARS404 qRT-PCR, amplicon downstream ARS404 ACS. 5' primes 288 bp after SSB1 STOP, pair with DL3763. |
| | DL3767 | CTTTTTAAACTAATATA CACATTTTAGCAGATGCG | ARS404 qRT-PCR, amplicon upstream ARS404 ACS. 5' primes 23 bp after HO STOP, pair with DL3768. |
| | DL3768 | GATGCTGTCCG CGGGCCTCATAAG | ARS404 qRT-PCR, amplicon upstream ARS404 ACS. 5' primes 60 bp before HO STOP, pair with DL3767. |
| | DL3823 | GGCACTATGCTTTTT AAAATTTTGTTTATACTCAATTTCG | ARS1004 qRT-PCR, amplicon upstream ARS1004 ACS. 5' anneals 80 bp after REE1 STOP |
| | DL3824 | GCCCAGTATTTTGTT AACTGTATGGATTGTACTAG | ARS1004 qRT-PCR, amplicon upstream ARS1004 ACS. 5' anneals 170 bp after REE1 STOP |
| | DL3827 | GTGTTTTAAGATA AAGTGACGAAAGTTAGGGTG | ARS1004 qRT-PCR, amplicon downstream ARS1004 ACS. 5' anneals 228 bp after REE1 STOP |
| | DL3828 | CATCATAAGTACTAATTA CCACGAATTCAATAATTAGTAAATAC | ARS1004 qRT-PCR, amplicon downstream ARS1004 ACS. 5' anneals 318 bp after REE1 STOP |
| | DL187 | ACACActaaattaccggatc aattcgggggatccAT GAACGACCAAACGCAATT | Cloning of *HSP104* in pCM188, fwd. |
| | DL189 | catgatgcggccctcctgcagggc cctagcggccgcTTAATCTAGGTCATCATCAA | Cloning of *HSP104* in pCM188, rev. |
| | DL1124 | taatgaggacagtatggaaatt gatgatgacctagattaa TTTAATATAGTGTGATTTTT | Cloning of *HSP104* 3' UTR in pCM188-*HSP104*, fwd. |
| | DL1125 | ATTACATGATGCGGCCCTC CTGCAGGGCCCTAGCGGCCGCTT TAACATGATTTGGTAGTC | Cloning of *HSP104* 3' UTR in pCM188-*HSP104*, fwd. |
| | DL4026 | CGTTTATTCCCTT GTTTGATTCAGAAGCAG | ARS1 KO in pDL214 by overlapping PCRs, Fwd. Anneals 236 bp after pDL214's *URA3* STOP. To be used for both 1 st and 2nd step of the reaction. During 1 st step, use it in combination with DL4027. During 2nd step, use it in combination with DL4030. |

*Table 1 continued on next page*

| Oligonucleotides | DL4027 | GCTAGCAAGAATCGGC TCGGGGCTCTCTTGCCTTCCAAC | ARS1 KO in pDL214 by overlapping PCRs, Rev. Anneals 334 bp after pDL214's *URA3* STOP. To be used during 1 st step in combination with DL4026. |
|---|---|---|---|
| | DL4029 | CAAGAGAGCCCCGAGC CGATTCTTGCTAGCCTTTTCTC | ARS1 KO in pDL214 by overlapping PCRs, Fwd. Anneals 746 bp after pDL214's *URA3* STOP. To be used during 1 st step in combination with DL4030. |
| | DL4030 | GATTACGAGG ATACGGAGAGAGG | ARS1 KO in pDL214 by overlapping PCRs, Rev. Anneal s 843 bp after pDL214's *URA3* STOP. To be used for both 1 st and 2nd step of the reaction. During 1 st step, use it in combination with DL 4029. During 2nd step, use it in combination with DL4026. |
| | DL4032 | GTGAAGGAGCAT GTTCGGCACAC | ARS1 KO in pDL214 by o verlapping PCRs, Rev sequencing primer. Anneals 1157 bp after pDL214's *URA3* STOP. |
| | DL4000 | TTCAAATGTACAGTAACTAT CAAAACCATT ATTGTAGTACCCGTA TTCTAATAATGAGCAAAAGAG CTCACATTTTAACG | Reverse ARS1206 orientation i n pDL214, Fwd. Bears 55 bp of homology with ARS1206 3' end (+320 to+375 after *HSP104* STOP ) followed by 25 bp of homology to 5' of T-rich predicted ACS (+102 to+127 after *HSP104* STOP). Pair with DL4001. |
| | DL4001 | TATATATAATTAATAAAACTAA TGGAATTTGTT TAATTGAACTTGACAC CCGAGCGGACC AATCCGCGTGTG TTTTATAC | Reverse ARS1206 orientation in pDL214, Rev. Bears 55 bp of homology with ARS1206 5' end (+51 to+106 after *HSP104* STOP) followed by 25 bp of homology with 3' end of ARS1206 (+295 to+320 after *HSP104* STOP). Pair with DL4000. |
| | DL4061 | ATTATTAGAATACGGGTACTAC | Reverse ARS1206 orientation in pDL214, extension of homology region downstream ARS1206, Fwd. Primes 134 bp upstream *CYC1* terminator. Pair with M13 reverse (DL2163). |
| | DL2163 | caggaaacagctatgac | Reverse ARS1206 orientation in pDL214, extension of homology region downstream ARS1206, Rev. |
| | DL4066 | GCTCGGGTGTCA AGTTCAATTAAAC | Reverse ARS1206 orientation in pDL214, extension of homology region upstream ARS1206, Rev. Primes 106 bp downstream *HSP104* STOP. Pair with DL530. |
| | DL530 | GTTGAATTTA ACTCAAGAGGC | Reverse ARS1206 orientation in pDL214, extension of homology region upstream ARS1206, Fwd. Anneals 2409–2429 in *HSP104*. |

*Table 1 continued on next page*

| Oligonucleotides | DL3986 | gctgaagaatg tctggaagttctacc | Reverse ARS1206 orientation in pDL214, Fwd sequencing primer annealing 108 bp before *HSP104* STOP. |
|---|---|---|---|
| | DL163 | acattttcatcacgagatttaccc | RNase H cleavage assay. *HSP104*, antisense, position 2606–2583 from *HSP104* ATG. |
| | DL164 | ttatcgtcatcacct aacgtgtcagcccta tagtagcttcgtg atttggtagaacttcc | RNase H cleavage assay. *HSP104* Northern oligonucleotide probe, antisense, position 2718–2631 from *HSP104* ATG. |
| | DL473 | TTTTTTTTTTT TTTTTTTTT | RNase H cleavage assay. Poly(dT) oligonucleotide |
| | DL3991 | GATTTGACGTCCAG TGGACTTTTTTGTCC | RNase H cleavage assay, test *HSP104* readthrough on pDL905, antisense, position 2923–2895 from *HSP104* ATG |
| | DL3994 | GGAAGTAATAAGTGAA GGTTAAATCTGGACC | RNase H cleavage assay, t est *HSP104* readthrough on pDL907, antisense, position 2909–2879 from *HSP104* ATG |

| Plasmids | Name | Features | Reference |
|---|---|---|---|
| | pDL454 | PTETOFF-HSP104::Reb1BS::HSP104, PGAL1-CUP1, 2μ, URA3 | Colin et al. **Colin et al., 2014** |
| | pDL551 | PTETOFF-HSP104::Reb1BS(−)::HSP104, PGAL1-CUP1, 2μ, URA3 | |
| | pDL790 | PTETOFF-HSP104::ARS305_548 bp::HSP104, PGAL1-CUP1, 2μ, URA3 | This study |
| | pDL793 | PTETOFF-HSP104::ARS305(A−)_548 bp::HSP104, PGAL1-CUP1, 2μ, URA3 | |
| | pDL909 | PTETOFF-HSP104::ARS305_175 bp::HSP104, PGAL1-CUP1, 2μ, URA3 | |
| | pDL910 | PTETOFF-HSP104::ARS305(A−)_175 bp::HSP104, PGAL1-CUP1, 2μ, URA3 | |
| | pDL911 | PTETOFF-HSP104::ARS305(B1−)_175 bp::HSP104, PGAL1-CUP1, 2μ, URA3 | |
| | pDL912 | PTETOFF-HSP104::ARS305(B4−)_175 bp::HSP104, PGAL1-CUP1, 2μ, URA3 | |
| | pDL913 | PTETOFF-HSP104::ARS305(B1−B4−)_175 bp::HSP104, PGAL1-CUP1, 2μ, URA3 | |
| | pDL30 | PTETOFF-HSP104, ARS1, CEN4, URA3 | Libri laboratory |
| | pDL214 | PTETOFF-HSP104, ARS1206, ARS1, CEN4, URA3 | |
| | pDL905 | PTETOFF-HSP104, ARS1206, Δars1, CEN4, URA3 | This study |
| | pDL907 | PTETOFF-HSP104, 6021sra, Δars1, CEN4, URA3 | |

DOI: https://doi.org/10.7554/eLife.40802.008

**Table 2.** Coordinates of primary and secondary ACSs used in this study.

| | | | Proposed primary ACS (Nieduszynski et al., 2006) | | | | | Putative secondary ACS (this study) | | | | | | |
|---|---|---|---|---|---|---|---|---|---|---|---|---|---|---|
| ID | Chromosome | Strand | Start | End | Match | Score | Chromosome | Strand | Start | End | Match | Score | Protected length (nt) | |
| 1 | chrI | + | 31001 | 31018 | TATTTTTAAGTTTTGTT | 0.974909231 | chrI | - | 31190 | 31173 | GTATAATATTTTTAGTT | 0.87301127 | 189 |
| 2 | chrI | - | 70431 | 70414 | ATTTTTATGTTTAGAA | 0.949548431 | chrI | + | 70251 | 70268 | ACTATCAATGTTTATC | 0.818662772 | 180 |
| 3 | chrI | - | 124526 | 124509 | ATTTTTATATTTAAGT | 0.939615332 | chrI | + | 124412 | 124429 | GTTTTCTCTATTTAAAT | 0.76163459 | 114 |
| 4 | chrI | + | 159951 | 159968 | TTTATTTATATTTAGTG | 0.951660057 | chrI | - | 160108 | 160091 | ATATAGCATAATTACTT | 0.796339361 | 157 |
| 5 | chrI | + | 176234 | 176251 | TCTTTTTATGTTTCTT | 0.936946746 | chrI | - | 176333 | 176316 | TAAATATGTGTTTATTA | 0.816621821 | 99 |
| 6 | chrII | + | 28984 | 29001 | TCACTCTATCTTTTTA | 0.78989004 | chrII | - | 29092 | 29075 | TATAACAAAAATTGGTC | 0.767973746 | 108 |
| 7 | chrII | - | 63376 | 63359 | TTTTTTAATTTTTGTC | 0.934538928 | chrII | + | 63256 | 63273 | TAAAAATTTGTTTTCTT | 0.843331211 | 120 |
| 8 | chrII | - | 170228 | 170211 | CCAGTGAACGCTTAAAA | 0.646819795 | chrII | + | 170126 | 170143 | CTTTGCTACGGATTTCTT | 0.76319826 | 102 |
| 9 | chrII | - | 198382 | 198365 | AACTTCAAAGTACATTG | 0.673812699 | chrII | + | 198228 | 198245 | ATTATAGACTTTCATTC | 0.772245255 | 154 |
| 10 | chrII | - | 237832 | 237815 | AAGGTACATAGCGATTT | 0.628400298 | chrII | + | 237685 | 237702 | TTATTAAAGGGTTTGGA | 0.774836934 | 147 |
| 11 | chrII | - | 255040 | 255023 | AGGTAGAAGAGTTACGG | 0.617416402 | chrII | + | 254892 | 254909 | TGATTTTCATTTTACT | 0.841326164 | 148 |
| 12 | chrII | + | 326149 | 326166 | CTATCGAAACTTTGTT | 0.748562634 | chrII | - | 326273 | 326256 | CTTTTAATAGTTTAGGT | 0.860235002 | 124 |
| 13 | chrII | - | 408006 | 407989 | TAGGAAAATATATAGAG | 0.708025047 | chrII | + | 407871 | 407888 | ATATTTAAAGAGTTGAA | 0.77590664 | 135 |
| 14 | chrII | - | 417974 | 417957 | TGTAGAAATGTCTAGCG | 0.67916971 | chrII | + | 417844 | 417861 | AAATTTAATATTTTGA | 0.912902242 | 130 |
| 15 | chrII | - | 486855 | 486838 | GAAGTCCTCTTCTTCGC | 0.639951668 | chrII | + | 486735 | 486752 | ATTAATTATGTTTTTCC | 0.89533109 | 120 |
| 16 | chrII | + | 622713 | 622730 | TATATAGAAAGTTGCTT | 0.760778109 | chrII | - | 622866 | 622849 | TTTTTGTACGTTTTTT | 0.907808059 | 153 |
| 17 | chrII | + | 704289 | 704306 | CTACCAAAAGTGTACCG | 0.581803503 | chrII | - | 704455 | 704438 | AATGTTTTTTTTTTT | 0.897759223 | 166 |
| 18 | chrII | - | 741746 | 741729 | CGAAAAGATATGTGGGA | 0.64946824 | chrII | + | 741628 | 741645 | TAAGATCAAGTTTGGTA | 0.824844021 | 118 |
| 19 | chrII | + | 757441 | 757458 | TAAATCTAAGATAGCTG | 0.682422088 | chrII | - | 757613 | 757596 | GTTATATAAGTATACGT | 0.779064174 | 172 |
| 20 | chrII | + | 792164 | 792181 | TATTTCATGGTTTTTAG | 0.736834685 | chrII | - | 792287 | 792270 | CTTTTTAAAATTCATTG | 0.834945362 | 123 |
| 21 | chrIII | + | 11254 | 11271 | TTTTTTTATGTTTTTTT | 0.985847127 | chrIII | - | 11400 | 11383 | GTTGAATTTGGTTAGAT | 0.782826917 | 146 |
| 22 | chrIII | - | 39591 | 39574 | TTTTTATATGTTTTGTT | 0.963617028 | chrIII | + | 39476 | 39493 | TTATTTTTTATTTACTT | 0.914777509 | 115 |
| 23 | chrIII | + | 74518 | 74535 | TGTATTTATATTTATTT | 0.944792175 | chrIII | - | 74682 | 74665 | GAGATCTTAATTTATCT | 0.770457519 | 164 |
| 24 | chrIII | - | 108972 | 108955 | TTTATTTATGTTTTCTT | 0.960865701 | chrIII | + | 108832 | 108849 | TAGAAATATGTTGAGTT | 0.795588546 | 140 |
| 25 | chrIII | + | 132036 | 132053 | TTTGTACATTGTTTATA | 0.792015393 | chrIII | - | 132155 | 132138 | CTTTTATATGTTTAAAT | 0.885104513 | 119 |
| 26 | chrIII | + | 166650 | 166667 | GTTTTATTCCATTATTT | 0.81768767 | chrIII | - | 166768 | 166751 | ATTATTTACATTTACGA | 0.903103359 | 118 |
| 27 | chrIII | + | 194302 | 194319 | CTACTGCAAATTTTTAC | 0.730959168 | chrIII | - | 194402 | 194385 | TGTAATTACATTTCTTA | 0.79211775 | 100 |
| 28 | chrIII | - | 197559 | 197542 | AATATTCATGTTTAGTA | 0.934784063 | chrIII | + | 197415 | 197432 | ATCTTAAAACCTTTTAG | 0.797219912 | 144 |
| 29 | chrIII | + | 224856 | 224873 | TCAGTTTTTTTATGTT | 0.78153895 | chrIII | - | 224956 | 224939 | TTTATTTTGTTTGTTT | 0.899494022 | 100 |
| 30 | chrIII | - | 273030 | 273013 | TTTTTTCAAATTTAGTT | 0.94325972 | chrIII | + | 272904 | 272921 | TTTATTCAAAATTTTTC | 0.870692365 | 126 |
| 31 | chrIII | + | 292584 | 292601 | TATATATATATTTATTT | 0.933162383 | chrIII | - | 292695 | 292678 | TATAATAACATTTTTTA | 0.881496782 | 111 |
| 32 | chrIII | + | 315872 | 315889 | TGTATATAAATTAAGTG | 0.777607317 | chrIII | - | 315979 | 315962 | CATTTTAATATCTATAT | 0.829435873 | 107 |
| 33 | chrIV | - | 15681 | 15664 | ATTTTTACGTTTTCTC | 0.928797007 | chrIV | + | 15525 | 15542 | TAAATTCTAAGTTATTC | 0.806599978 | 156 |
| 34 | chrIV | - | 86123 | 86106 | GATTTTTATGTTTGGGC | 0.907628171 | chrIV | + | 85996 | 86013 | CTTTATAAAGATTTTAT | 0.843543061 | 127 |

*Table 2 continued on next page*

Table 2 continued

| | Proposed primary ACS (Nieduszynski et al., 2006) | | | | | | Putative secondary ACS (this study) | | | | | | |
|---|---|---|---|---|---|---|---|---|---|---|---|---|---|
| ID | Chromosome | Strand | Start | End | Match | Score | Chromosome | Strand | Start | End | Match | Score | Protected length (nt) |
| 35 | chrIV | + | 123677 | 123694 | TGTTTCACTTTGTGTT | 0.820618605 | chrIV | - | 123793 | 123776 | TTAATATATATTTAGTT | 0.9347773 | 116 |
| 36 | chrIV | - | 212592 | 212575 | TTTTTTTATATTTTGTT | 0.991320747 | chrIV | + | 212441 | 212458 | TTTTTTTTTTTTTTTT | 0.926463613 | 151 |
| 37 | chrIV | + | 253839 | 253856 | ATTTTTATAGTTTTGC | 0.901024131 | chrIV | - | 253948 | 253931 | TAATTTATCTTTAGAT | 0.940018266 | 109 |
| 38 | chrIV | - | 329742 | 329725 | GATTTTATTTTTTGT | 0.930581986 | chrIV | + | 329601 | 329618 | TATTATTATTATTATTC | 0.884653435 | 141 |
| 39 | chrIV | + | 408134 | 408151 | TTATATTATATTTAGCG | 0.896228674 | chrIV | - | 408291 | 408274 | TTATTACATATTTTGT | 0.898263462 | 157 |
| 40 | chrIV | - | 484039 | 484022 | TTTTTTTATATTTATGT | 0.972409126 | chrIV | + | 483896 | 483913 | TTGTTTGTTCATTTCTT | 0.792451309 | 143 |
| 41 | chrIV | - | 505522 | 505505 | TTTTTTTATATTTTGC | 0.95203234 | chrIV | + | 505345 | 505362 | CCTTTTCACGTTTTGC | 0.864843823 | 177 |
| 42 | chrIV | - | 555401 | 555384 | AAAGTTTATGTTTTTTC | 0.925775335 | chrIV | + | 555290 | 555307 | ATAAATGTTGTTTTTT | 0.835510567 | 111 |
| 43 | chrIV | - | 567681 | 567664 | TTTTTTTATGTTTTGAG | 0.946669447 | chrIV | + | 567572 | 567589 | ACTTTTAATTTTTTTTT | 0.905571442 | 109 |
| 44 | chrIV | - | 640068 | 640051 | TTTTTAAAGTTTTGGT | 0.951500543 | chrIV | + | 639918 | 639935 | CTATAATATATTTATTC | 0.86149187 | 150 |
| 45 | chrIV | + | 702928 | 702945 | AAAATAATTAAATGTTTT | 0.737939741 | chrIV | - | 703030 | 703013 | TGATTTAAAAATTCTGTA | 0.83908476 | 102 |
| 46 | chrIV | + | 748452 | 748469 | AAATTAATTGATTAATT | 0.822458971 | chrIV | - | 748585 | 748568 | TTTTTTAATATTTAATA | 0.915446997 | 133 |
| 47 | chrIV | - | 753339 | 753322 | TTTTTTTACATTTTGCT | 0.953908195 | chrIV | + | 753221 | 753238 | AAACTTATTTTTAAGC | 0.78950557 | 118 |
| 48 | chrIV | + | 806097 | 806114 | CTCTTCCAAATTTTTAA | 0.777746734 | chrIV | - | 806256 | 806239 | TCATATCCTGTTTAAA | 0.722790604 | 159 |
| 49 | chrIV | + | 913859 | 913876 | TTTTTTATTTTTATAT | 0.943491396 | chrIV | - | 913957 | 913940 | ACAATTTTGTTTATTT | 0.885371567 | 98 |
| 50 | chrIV | + | 921736 | 921753 | TCTTTAATCGATTTTAA | 0.773941597 | chrIV | - | 921840 | 921823 | TTTGTTTATTTTTTTT | 0.943438157 | 104 |
| 51 | chrIV | - | 1016854 | 1016837 | TTTGTTTACGTTTTGGA | 0.934312886 | chrIV | + | 1016682 | 1016699 | AGAATTCATTTTAATCT | 0.772819262 | 172 |
| 52 | chrIV | + | 1057886 | 1057903 | TTCTTTTATTATTTTTT | 0.899933367 | chrIV | - | 1058017 | 1058000 | AAAGTGAATTTTTTTGT | 0.837029199 | 131 |
| 53 | chrIV | - | 1110139 | 1110122 | TTTTTTTATATTTTTAT | 0.956467815 | chrIV | + | 1109960 | 1109977 | GAATTCTTCATTTAGAT | 0.824896005 | 179 |
| 54 | chrIV | - | 1159452 | 1159435 | CTTTTCTAAGCTTTGAA | 0.769370807 | chrIV | + | 1159286 | 1159303 | ATAATTAAATTTTTTGA | 0.889208627 | 166 |
| 55 | chrIV | - | 1166166 | 1166149 | TCGGAATATATATTCTT | 0.763125812 | chrIV | + | 1166064 | 1166081 | CTTAATAAATTTTGTA | 0.854045557 | 102 |
| 56 | chrIV | + | 1240920 | 1240937 | CTTCTTGAAAATTGATT | 0.771311686 | chrIV | - | 1241096 | 1241079 | TTTATAAAAAATTATAT | 0.871453601 | 176 |
| 57 | chrIV | + | 1276271 | 1276288 | TTCGTTTTCTTTTTCTC | 0.82062871 | chrIV | - | 1276405 | 1276388 | CAAATATATATTGATCA | 0.767679431 | 134 |
| 58 | chrIV | - | 1302763 | 1302746 | TATATATTTAGTTAATG | 0.795859241 | chrIV | + | 1302616 | 1302633 | GAGTTTTACGTATTCTT | 0.80224896 | 147 |
| 59 | chrIV | + | 1404323 | 1404340 | TAAAATCATTTTCTTT | 0.829710275 | chrIV | - | 1404511 | 1404494 | AGGATTCTTTATTACGT | 0.774058834 | 188 |
| 60 | chrIV | + | 1461890 | 1461907 | GAGTAACTTCTTGTCGG | 0.624436491 | chrIV | - | 1462038 | 1462021 | AACATTAATTGTTGTTA | 0.790149896 | 148 |
| 61 | chrIV | - | 1487098 | 1487081 | TTAAATTTAGTTTTTT | 0.870549799 | chrIV | + | 1486965 | 1486982 | CCAATACACATGATTGGAT | 0.773138313 | 133 |
| 62 | chrV | - | 59469 | 59452 | AATATTTACATTTTGAT | 0.935717414 | chrV | + | 59363 | 59380 | TTTTTTTTTCTTTTTT | 0.922560213 | 106 |
| 63 | chrV | + | 94055 | 94072 | CAAGTTTATATTTTGTT | 0.938620288 | chrV | - | 94173 | 94156 | TATGTTTAATTATATTG | 0.79888376 | 118 |
| 64 | chrV | - | 145714 | 145697 | CAGTTTTTGTTTAGTT | 0.906995194 | chrV | + | 145608 | 145625 | TTATATAAATATTTTAGG | 0.854409653 | 106 |
| 65 | chrV | - | 173808 | 173791 | TAATTTTATATTTTGCC | 0.93759113 | chrV | + | 173704 | 173721 | TATTTATACTTTTACGG | 0.861582181 | 104 |
| 66 | chrV | + | 212455 | 212472 | TAAAATTATGTTTAGGT | 0.938368393 | chrV | - | 212555 | 212538 | CGTATACTTTTTTGTG | 0.794230687 | 100 |
| 67 | chrV | + | 287567 | 287584 | TTTATTTATGTTTTGTT | 0.988690479 | chrV | - | 287761 | 287744 | CTTTGTTATCTTGTGAA | 0.729422588 | 194 |
| 68 | chrV | + | 353586 | 353603 | AATATTTACTTTTTGGT | 0.936542643 | chrV | - | 353774 | 353757 | TTGAATTATGCTTATGT | 0.812386986 | 188 |

Table 2 continued

| | Proposed primary ACS (Nieduszynski et al., 2006) | | | | | | Putative secondary ACS (this study) | | | | | | |
|---|---|---|---|---|---|---|---|---|---|---|---|---|---|
| ID | Chromosome | Strand | Start | End | Match | Score | Chromosome | Strand | Start | End | Match | Score | Protected length (nt) |
| 69 | chrV | - | 406906 | 406889 | TTTTTTTATATATAGTC | 0.88971164 | chrV | + | 406734 | 406751 | GTAATTTATGATTAAATC | 0.864888268 | 172 |
| 70 | chrV | - | 439105 | 439088 | ATTTTTTAAGTTTTGCG | 0.915882066 | chrV | + | 438997 | 439014 | GGTATTCTTCTTTTTCT | 0.814453982 | 108 |
| 71 | chrV | + | 549589 | 549606 | TATTATTAATATCTTGT | 0.818517794 | chrV | - | 549686 | 549669 | TAATTTAATATTTTTTT | 0.948482332 | 97 |
| 72 | chrV | - | 167738 | 167721 | TATATTTATATTTCGT | 0.945765544 | chrVI | + | 167551 | 167568 | AATATTTAAATATAAGT | 0.814242246 | 187 |
| 73 | chrVI | + | 199397 | 199414 | TTATTCGAGCTTTGTC | 0.737504399 | chrVI | - | 199507 | 199490 | ATCCATAATATTTACCT | 0.801830214 | 110 |
| 74 | chrVI | + | 216470 | 216487 | CATTTCTATTTTTTTT | 0.890722071 | chrVI | - | 216600 | 216583 | TAATGTGATGGTTAGTT | 0.802062704 | 130 |
| 75 | chrVI | - | 256383 | 256366 | TTTATGTTTTTCCGGA | 0.701845209 | chrVI | + | 256263 | 256280 | AAAAATTCCGATCTTGT | 0.72753389 | 120 |
| 76 | chrVII | - | 64458 | 64441 | ATTTTAATATTTTGTT | 0.966859378 | chrVII | + | 64357 | 64374 | TATTGTATATTTAGTT | 0.901272249 | 101 |
| 77 | chrVII | + | 112124 | 112141 | ATTTTATACGTTTATGT | 0.921703978 | chrVII | - | 112271 | 112254 | ATAGTTTTTTTTATGC | 0.861155565 | 147 |
| 78 | chrVII | + | 163235 | 163252 | TCATTTTATAATTTGTT | 0.916233817 | chrVII | - | 163378 | 163361 | GTAATATATGATTAGAA | 0.844307348 | 143 |
| 79 | chrVII | + | 203971 | 203988 | ATTTTTATATTATTA | 0.950625858 | chrVII | - | 204165 | 204148 | CATTTAAACTCTATAT | 0.7880761 | 194 |
| 80 | chrVII | + | 286003 | 286020 | TTTATTTACTTTTAGTC | 0.933155022 | chrVII | - | 286153 | 286136 | CTAGTAATCTTTCAGTC | 0.747097252 | 150 |
| 81 | chrVII | - | 352863 | 352846 | TTTAATTACGTTTAGTT | 0.942276914 | chrVII | + | 352758 | 352775 | TACTTTTATGATTCATT | 0.812763403 | 105 |
| 82 | chrVII | - | 388846 | 388829 | TTTATTTAACCTTTGTT | 0.939702794 | chrVII | + | 388738 | 388755 | TTAGTTCTCATTTATAA | 0.82432824 | 108 |
| 83 | chrVII | - | 421280 | 421263 | ATAAATTATTGTTTAGT | 0.826708937 | chrVII | + | 421176 | 421193 | CTATTTCAAAATTGTTT | 0.859366438 | 104 |
| 84 | chrVII | - | 485110 | 485093 | TTTATTTATGTTTTGCC | 0.947613634 | chrVII | + | 484978 | 484995 | AATTATCAAGTTTTTCT | 0.875154553 | 132 |
| 85 | chrVII | - | 508907 | 508890 | CATTTAATGTTTGGTT | 0.923555282 | chrVII | + | 508801 | 508818 | ATCTTTATCTTTTATC | 0.872797056 | 106 |
| 86 | chrVII | - | 568660 | 568643 | AGTATTATATTTAGCC | 0.909439604 | chrVII | + | 568509 | 568526 | GTCATTCATGATTTATT | 0.834093344 | 151 |
| 87 | chrVII | + | 574700 | 574717 | AGTATTTATGTTTTGTC | 0.937749085 | chrVII | - | 574854 | 574837 | TATACTCATATTTTGGC | 0.338055118 | 154 |
| 88 | chrVII | - | 660000 | 659983 | ATATTTTATGTTTACTT | 0.952756007 | chrVII | + | 659904 | 659921 | TTGTTTTTTATTGTTT | 0.823819951 | 96 |
| 89 | chrVII | + | 715314 | 715331 | TTTGTTTATATTTGTT | 0.970567449 | chrVII | - | 715431 | 715414 | AATCTTTAACTTGTGAT | 0.779912848 | 117 |
| 90 | chrVII | + | 778013 | 778030 | CTTTTTTACCTTTTGTT | 0.938434047 | chrVII | - | 778193 | 778176 | AGTGTTTATATTTATTT | 0.926919799 | 180 |
| 91 | chrVII | - | 834664 | 834647 | TTGTATATAGTTTAGTT | 0.854509956 | chrVII | + | 834549 | 834566 | GGTTTTTAACTTTTCCC | 0.830646453 | 115 |
| 92 | chrVII | + | 888412 | 888429 | TATTTTAATATTTAGTT | 0.973625821 | chrVII | - | 888567 | 888550 | TTTATATATATATATTC | 0.823335292 | 155 |
| 93 | chrVII | - | 977904 | 977887 | TTTTTAATTTTTTAT | 0.925318963 | chrVII | + | 977810 | 977827 | TTTTTTAATGATTTTT | 0.806000942 | 94 |
| 94 | chrVII | + | 999468 | 999485 | CTTTTTTACTTTTGGG | 0.904948204 | chrVII | - | 999575 | 999558 | TATTTTTTTTTTTTT | 0.925871289 | 107 |
| 95 | chrVIII | - | 7755 | 7738 | TATTTTTATATTTAGGT | 0.984899843 | chrVIII | + | 7618 | 7635 | CTTGTTTATTATTATTA | 0.875022851 | 137 |
| 96 | chrVIII | + | 64302 | 64319 | TAATTTTAATTTTAGTT | 0.942262943 | chrVIII | - | 64434 | 64417 | ATTCTTTATATTTATTT | 0.922675429 | 132 |
| 97 | chrVIII | - | 133538 | 133521 | TATTTTAACATTTAGTT | 0.959052991 | chrVIII | + | 133406 | 133423 | TTCTTTATGTGTATGC | 0.834208883 | 132 |
| 98 | chrVIII | + | 168597 | 168614 | TTGTGTCATATTTAGAC | 0.799695233 | chrVIII | - | 168793 | 168776 | TATATATATATATACGT | 0.820409776 | 196 |
| 99 | chrVIII | + | 245788 | 245805 | CTATTTTATGATTAGTT | 0.939777326 | chrVIII | - | 245940 | 245923 | CAATTCCAAATTTAGGC | 0.831524522 | 152 |
| 100 | chrVIII | - | 392260 | 392243 | TTTTTTCTTGAGTACTT | 0.788764838 | chrVIII | + | 392088 | 392105 | ATAATTTACATTAATAT | 0.821200767 | 172 |
| 101 | chrVIII | - | 447794 | 447777 | TATGTTTATGTTTTGTG | 0.947093715 | chrVIII | + | 447598 | 447615 | TTGCTTAATATTTTGCA | 0.846461752 | 196 |
| 102 | chrVIII | - | 501949 | 501932 | CGTTTATACATTTTGTT | 0.896794884 | chrVIII | + | 501752 | 501769 | ATATTTTACGGTTCTTT | 0.824337524 | 197 |

Table 2 continued on next page

*Table 2 continued*

| | Proposed primary ACS (Nieduszynski et al., 2006) | | | | | | Putative secondary ACS (this study) | | | | | | |
|---|---|---|---|---|---|---|---|---|---|---|---|---|---|
| ID | Chromosome | Strand | Start | End | Match | Score | Chromosome | Strand | Start | End | Match | Score | Protected length (nt) |
| 103 | chrVIII | + | 556140 | 556157 | AATTTTACGTTTAGGT | 0.969507836 | chrVIII | - | 556301 | 556284 | CATTTTAAATATCTATAT | 0.829435873 | 161 |
| 104 | chrIX | - | 105966 | 105949 | ATTATTCATGTTTTCTT | 0.92780469 | chrIX | + | 105812 | 105829 | AATAATAATAATAATGG | 0.754881026 | 154 |
| 105 | chrIX | - | 136290 | 136273 | GCAGTTTATGTTTTGTT | 0.905839044 | chrIX | + | 136160 | 136177 | GATATCTATATTTTATA | 0.840946348 | 130 |
| 106 | chrIX | + | 175173 | 175190 | ATGTTTTATGTTTTGTC | 0.936874196 | chrIX | - | 175339 | 175322 | CAATTTCAAATTTAAAA | 0.82970169 | 166 |
| 107 | chrIX | + | 214735 | 214752 | TTAATTTATGTTTTGTA | 0.95530712 | chrIX | - | 214909 | 214892 | TGTTTTATATATTCGT | 0.841209426 | 174 |
| 108 | chrIX | - | 245882 | 245865 | TTTTTTAATGTTTTGTC | 0.962520612 | chrIX | + | 245773 | 245790 | CCTTAAAAAGGTCTCAC | 0.67119524 | 109 |
| 109 | chrIX | - | 247754 | 247737 | TTTTTTAATGTTTTGTC | 0.962520612 | chrIX | + | 247631 | 247648 | TACATTTCTCTTTTTT | 0.823299168 | 123 |
| 110 | chrIX | - | 342031 | 342014 | TTTTTTAATGTTTAGCT | 0.961127508 | chrIX | + | 341853 | 341870 | TAAGGTCTTGTTTGTTT | 0.760099392 | 178 |
| 111 | chrIX | + | 357225 | 357242 | AATTTTTATATTTTGTT | 0.983369656 | chrIX | - | 357356 | 357339 | TATTTATAGATTTTTCT | 0.83281607 | 131 |
| 112 | chrIX | - | 412003 | 411986 | AATTTTAATGTTTTGTC | 0.954569521 | chrIX | + | 411895 | 411912 | AAGGTATAAATGTAGTT | 0.778441725 | 108 |
| 113 | chrX | - | 7731 | 7714 | TATTTTTATGTTTAGGT | 0.992509265 | chrX | + | 7570 | 7587 | CATTTTAAATATCTATAT | 0.829435873 | 161 |
| 114 | chrX | - | 67714 | 67697 | CTTTTTTATTTTTTTT | 0.944897067 | chrX | + | 67593 | 67610 | AAAATTAATAAATTTCC | 0.769826733 | 121 |
| 115 | chrX | + | 99498 | 99515 | TTTTTTAATTTTTTTTT | 0.947088854 | chrX | - | 99625 | 99608 | TTTATTTATGTTTTGTT | 0.988690479 | 127 |
| 116 | chrX | + | 298616 | 298633 | TGACTCTAACTCCAGTT | 0.666661983 | chrX | - | 298725 | 298708 | CTAATAAAACTTTTTCC | 0.801772328 | 109 |
| 117 | chrX | + | 337049 | 337066 | CTTAAATAAGGTGAAGA | 0.678459288 | chrX | - | 337193 | 337176 | CTCTTGCTTGTTTAGTT | 0.819488866 | 144 |
| 118 | chrX | + | 374633 | 374650 | AATTACTACAATTTTCG | 0.788091986 | chrX | - | 374774 | 374757 | GAAATTTACATTTATTT | 0.914653679 | 141 |
| 119 | chrX | - | 375586 | 375569 | TTAGTGCAAAATATGAG | 0.674815863 | chrX | + | 375403 | 375420 | TTCTTTAAACTTTTTGA | 0.856145267 | 183 |
| 120 | chrX | - | 417088 | 417071 | TTGATGCACTATCATGA | 0.704755133 | chrX | + | 416918 | 416935 | GATTTCTATGTTCTCGA | 0.808544598 | 170 |
| 121 | chrX | + | 540294 | 540311 | GGGTAAAATGCGCTGTA | 0.572247037 | chrX | - | 540461 | 540444 | AAAAATTACTTCCAGTT | 0.755451504 | 167 |
| 122 | chrX | - | 612772 | 612755 | CACCAACAAATTGACAG | 0.600434727 | chrX | + | 612662 | 612679 | GGATTTCATAATTGTGG | 0.785437954 | 110 |
| 123 | chrX | - | 654253 | 654236 | TAAAGTTAACGTAACCA | 0.631991513 | chrX | + | 654127 | 654144 | TCAAAACTTGATTTGTT | 0.783019587 | 126 |
| 124 | chrX | + | 683708 | 683725 | CAGATAAAACAGACATAT | 0.624200951 | chrX | - | 683904 | 683887 | GTATTGTACATTTACCT | 0.826577659 | 196 |
| 125 | chrX | + | 711652 | 711669 | ATTTCTAATGCCTTGTG | 0.672178619 | chrX | - | 711852 | 711835 | TTTGTTCACTGTTAGTT | 0.872596683 | 200 |
| 126 | chrX | + | 729810 | 729827 | TAGTTGAATAATTCGTA | 0.742850129 | chrX | - | 729989 | 729972 | CGATTAAGCGTTTTGCC | 0.743397787 | 179 |
| 127 | chrX | - | 736901 | 736884 | CAATTGGAAAATTAGTG | 0.76415065 | chrX | + | 736789 | 736806 | TGTTTGGAGTGTTCAGGT | 0.744514544 | 112 |
| 128 | chrX | + | 744625 | 744642 | TAATTAGCACTTCTCCC | 0.637153506 | chrX | - | 744819 | 744802 | GTAATATAACTGTACTC | 0.72903611 | 194 |
| 129 | chrXI | - | 55866 | 55849 | TTCATTAATGTTTAGTT | 0.937267458 | chrXI | + | 55685 | 55702 | ATTTTTCATCTTTATTA | 0.906973964 | 181 |
| 130 | chrXI | + | 98384 | 98401 | TTTTTTTATGTTTAGTG | 0.969509169 | chrXI | - | 98530 | 98513 | GTACTTTATTTTTGGTT | 0.851436401 | 146 |
| 131 | chrXI | - | 153120 | 153103 | AATTTTTACAATTTGTC | 0.919552201 | chrXI | + | 152995 | 153012 | TAGTTATAAGATTATCT | 0.841554901 | 125 |
| 132 | chrXI | - | 196216 | 196199 | TTTTTTCATTTTTGTT | 0.951572253 | chrXI | + | 196020 | 196037 | TTTGCTCATTTTAAGT | 0.795946302 | 196 |
| 133 | chrXI | - | 213317 | 213300 | AGAGTTTGTCATTACCA | 0.719440701 | chrXI | + | 213207 | 213224 | ATTAATAATCTGTATTT | 0.803703635 | 110 |
| 134 | chrXI | - | 329497 | 329480 | GGTACTGAAAATTTCGGT | 0.675922258 | chrXI | + | 329388 | 329405 | AAAATTCTTGATGTGTT | 0.785345702 | 109 |
| 135 | chrXI | + | 388665 | 388682 | GGTGTTTAAGGGGTAAAT | 0.710373823 | chrXI | - | 388778 | 388761 | TTCGTTTTAGTTAGTA | 0.833546833 | 113 |
| 136 | chrXI | + | 416880 | 416897 | CGCGAGATCCATAGGCT | 0.528888624 | chrXI | - | 416990 | 416973 | TATATTCTTGATTGGAT | 0.835644767 | 110 |

Table 2 continued

| | Proposed primary ACS (Nieduszynski et al., 2006) | | | | | | Putative secondary ACS (this study) | | | | | | |
|---|---|---|---|---|---|---|---|---|---|---|---|---|---|
| ID | Chromosome | Strand | Start | End | Match | Score | Chromosome | Strand | Start | End | Match | Score | Protected length (nt) |
| 137 | chrXI | - | 447845 | 447828 | CACATACACATATTTAAC | 0.785193796 | chrXI | + | 447678 | 447695 | GTAATAAATATTCTCAT | 0.786845724 | 167 |
| 138 | chrXI | + | 516676 | 516693 | ACTTGTTATGGTTATGT | 0.80432569 | chrXI | - | 516825 | 516808 | CATAATTGCCTTTTCTT | 0.777169896 | 149 |
| 139 | chrXI | + | 581535 | 581552 | ACTATGTATCTTGCAGT | 0.639967512 | chrXI | - | 581699 | 581682 | TATTTTTTAATTATGC | 0.885914166 | 164 |
| 140 | chrXI | - | 612054 | 612037 | TTTGGATTCATCTAACG | 0.610536381 | chrXI | + | 611861 | 611878 | GAGAATGACGATTCCGT | 0.68160383 | 193 |
| 141 | chrXI | + | 642416 | 642433 | GGATGCGACATTTAACT | 0.658787349 | chrXI | - | 642546 | 642529 | CGCTTATATGTTGGTAT | 0.720382898 | 130 |
| 142 | chrXII | + | 91467 | 91484 | CATTTAACGTTTAGTT | 0.947368024 | chrXI | - | 91595 | 91578 | TCCTTTAAACTTTAGTT | 0.864360818 | 128 |
| 143 | chrXII | + | 156701 | 156718 | TGATTTTACTTTTTGGA | 0.897074392 | chrXII | - | 156822 | 156805 | TAAGATTACGTTTTTAA | 0.861864859 | 121 |
| 144 | chrXII | + | 231249 | 231266 | TTTGTTTATATTTTTGT | 0.950585996 | chrXII | - | 231358 | 231341 | GTTGTTTAGTTTTATTT | 0.830642974 | 109 |
| 145 | chrXII | - | 289420 | 289403 | AAAATTAATGTTTTGCT | 0.929806448 | chrXII | + | 289325 | 289342 | TATATCCTCTTTATAT | 0.811743224 | 95 |
| 146 | chrXII | - | 373327 | 373310 | TTTTTTATATTTTCTC | 0.944189014 | chrXII | + | 373227 | 373244 | TTCGATAAAGGTTTGTC | 0.807458273 | 100 |
| 147 | chrXII | - | 412852 | 412835 | ATGTTTTTGTTTTGTT | 0.918453308 | chrXII | + | 412678 | 412695 | GTTTTGTACCTTTAGCT | 0.848513235 | 174 |
| 148 | chrXII | - | 450659 | 450642 | TTTTTTTATATCTTGCT | 0.878438397 | chrXII | + | 450505 | 450522 | CGTTTTTATGTTTATTC | 0.924039943 | 154 |
| 149 | chrXII | - | 459090 | 459073 | ATTGTTTATGTTTTGTG | 0.940327272 | chrXII | + | 458995 | 459012 | CTATTCTATGTTTTCTT | 0.886167882 | 95 |
| 150 | chrXII | - | 513083 | 513066 | TTTATTTATGTTTTTGT | 0.968709027 | chrXII | + | 512958 | 512975 | ATTATAAACATTTTATA | 0.845822907 | 125 |
| 151 | chrXII | - | 603109 | 603092 | TTTTTAATGTTTATGT | 0.962915946 | chrXII | + | 602997 | 603014 | GTTTTTATCAGTTTCAT | 0.801484796 | 112 |
| 152 | chrXII | + | 659892 | 659909 | GCTTTTTATGTTTATTT | 0.92663958 | chrXII | - | 660003 | 659986 | AGTATTCATGTTTTACT | 0.871065837 | 111 |
| 153 | chrXII | - | 745115 | 745098 | TATCTTTATGTTTTGTT | 0.949064504 | chrXII | + | 745006 | 745023 | TCGTTCAAACTTTTGTC | 0.79040136 | 109 |
| 154 | chrXII | - | 794207 | 794190 | AAAGTTTAAGTTTAGTT | 0.935806549 | chrXII | + | 794096 | 794113 | TTTGATCATAATTATTT | 0.872143422 | 111 |
| 155 | chrXII | - | 888740 | 888723 | GTTTTTTATGTTTAGAT | 0.952111375 | chrXII | + | 888618 | 888635 | AATTTTTATAATTAATG | 0.88656275 | 122 |
| 156 | chrXII | + | 1007232 | 1007249 | ATGTTTCATATTTTTAT | 0.888016553 | chrXII | - | 1007338 | 1007321 | AAAATTTATAATTTAGT | 0.886785202 | 106 |
| 157 | chrXII | + | 1013789 | 1013806 | TTTTTTATGTTTTCTC | 0.951798435 | chrXII | - | 1013882 | 1013865 | AAACAGTACGTATTTTT | 0.715569985 | 93 |
| 158 | chrXII | - | 1024156 | 1024139 | CTTAATGATGTTTAGTT | 0.887516109 | chrXII | + | 1024017 | 1024034 | CTAGTTTTAATTATAT | 0.838833831 | 139 |
| 159 | chrXIII | + | 31766 | 31783 | GTAGTTTATTATTTAGTT | 0.89054401 | chrXIII | - | 31876 | 31859 | CATTAAAATAATTATAT | 0.824526619 | 110 |
| 160 | chrXIII | - | 94390 | 94373 | ATTAATTATATTTAGAT | 0.921181496 | chrXIII | + | 94266 | 94283 | ATGTTAAATATTTTATT | 0.857637919 | 124 |
| 161 | chrXIII | + | 137321 | 137338 | AATATTTATGTTTTTGTT | 0.980739388 | chrXIII | - | 137437 | 137420 | TTGTTATTTATTTTTGA | 0.841585149 | 116 |
| 162 | chrXIII | - | 184017 | 184000 | GTTATATATGGTTAGTT | 0.884678994 | chrXIII | + | 183864 | 183881 | ACATTAAAATATTTTGG | 0.834854862 | 153 |
| 163 | chrXIII | + | 263126 | 263143 | ATTTTTTATATTTTGTG | 0.953471148 | chrXIII | - | 263313 | 263296 | TATGTATATATTTATCT | 0.900878883 | 187 |
| 164 | chrXIII | + | 286846 | 286863 | ATTTTTCTTATTTAGTT | 0.921601724 | chrXIII | - | 286946 | 286929 | AGGATTTATGTTTTTTT | 0.908582747 | 100 |
| 165 | chrXIII | + | 371020 | 371037 | AATTTTATTGTTTAGTT | 0.937218464 | chrXIII | - | 371128 | 371111 | CACTTATATTTTTTAT | 0.851831461 | 108 |
| 166 | chrXIII | + | 468237 | 468254 | TTTTTTATTTTTTGTT | 0.977274497 | chrXIII | - | 468357 | 468340 | ATCATTTTTAATTAGTA | 0.851483278 | 120 |
| 167 | chrXIII | - | 535770 | 535753 | TTAATTATATTTTAGTT | 0.970090441 | chrXIII | + | 535662 | 535679 | AGTTGTTTTGTTTTTTT | 0.82595884 | 108 |
| 168 | chrXIII | + | 611318 | 611335 | ATTGTTTATGTTTATGT | 0.951906482 | chrXIII | - | 611459 | 611442 | ATTTGGCATCATTGTAT | 0.685281331 | 141 |
| 169 | chrXIII | + | 634521 | 634538 | TATTTTTACTATTTGTA | 0.910848762 | chrXIII | + | 634639 | 634622 | CAATTTTATGGTCATTT | 0.857274617 | 118 |
| 170 | chrXIII | + | 649362 | 649379 | TTATTTCATATTTTGTT | 0.953558055 | chrXIII | - | 649549 | 649532 | CTTACTAACAATTTCTC | 0.76251583 | 187 |

Table 2 continued on next page

Table 2 continued

| | Proposed primary ACS (Nieduszynski et al., 2006) | | | | | | Putative secondary ACS (this study) | | | | | | | |
|---|---|---|---|---|---|---|---|---|---|---|---|---|---|---|
| ID | Chromosome | Strand | Start | End | Match | Score | Chromosome | Strand | Start | End | Match | Score | Score | Protected length (nt) |
| 171 | chrXIII | - | 758417 | 758400 | AAATTTTATGTTTTTT | 0.965835588 | chrXIII | + | 758312 | 758329 | ACTTAGCGCGGTTTTT | 0.674331603 | | 105 |
| 172 | chrXIII | + | 772677 | 772694 | TTTTTTTACTATTACTT | 0.90600905 | chrXIII | - | 772820 | 772803 | AATTTATACAACTATAT | 0.778650456 | | 143 |
| 173 | chrXIII | + | 805162 | 805179 | TATTTTTGTATTTAGTC | 0.881724676 | chrXIII | - | 805312 | 805295 | TTTTTTTACCTTTTTCC | 0.903568549 | | 150 |
| 174 | chrXIII | + | 815391 | 815408 | AAATTCTATGTTTTGTT | 0.925333958 | chrXIII | - | 815493 | 815476 | ATTTTTTTTTTTTGGA | 0.903966564 | | 102 |
| 175 | chrXIII | - | 897976 | 897959 | TTTTTTATGTTTGGTT | 0.960544596 | chrXIII | + | 897881 | 897898 | TTATTTTATCATTTTCT | 0.88758988 | | 95 |
| 176 | chrXIV | - | 28654 | 28637 | TTTTTTTATTTTTAGGT | 0.971445917 | chrXIV | + | 28486 | 28503 | AAGTTAGATAAATTAGCG | 0.781498458 | | 168 |
| 177 | chrXIV | + | 61695 | 61712 | GTTTTAATGTTTTGTA | 0.934385921 | chrXIV | - | 61857 | 61840 | TTTATTTAAATTTGCC | 0.916575598 | | 162 |
| 178 | chrXIV | - | 89756 | 89739 | TATTTTTAAGTTTTGTT | 0.974909231 | chrXIV | + | 89644 | 89661 | CTACTTATAGTTTTCT | 0.805190002 | | 112 |
| 179 | chrXIV | - | 169748 | 169731 | TAATTTAACGTTTTGTT | 0.953532134 | chrXIV | + | 169589 | 169606 | TTTATATATATGTATGT | 0.835743836 | | 159 |
| 180 | chrXIV | - | 196225 | 196208 | TTTTTTAACTTTTAGCC | 0.904522219 | chrXIV | + | 196096 | 196113 | TTCGTAAAAATTTTTGC | 0.820044435 | | 129 |
| 181 | chrXIV | - | 250464 | 250447 | AATTTTTACGGTTTTT | 0.918603933 | chrXIV | + | 250330 | 250347 | GATAAACATATTCTTGT | 0.787486687 | | 134 |
| 182 | chrXIV | - | 280066 | 280049 | ATTATTTATGTTTTCT | 0.94647878 | chrXIV | + | 279948 | 279965 | ATAATAATTAATTAGTT | 0.843720251 | | 118 |
| 183 | chrXIV | + | 322003 | 322020 | TTTGTTTACGTTTAGGC | 0.937398674 | chrXIV | - | 322198 | 322181 | GTTATAAAATATTTATAA | 0.847440569 | | 195 |
| 184 | chrXIV | - | 412441 | 412424 | TTTTTTATATTTCTGC | 0.869234054 | chrXIV | + | 412299 | 412316 | CAACTTCTACATTACAT | 0.72789922 | | 142 |
| 185 | chrXIV | - | 449536 | 449519 | CATATTTACATTTAGCC | 0.905544669 | chrXIV | + | 449372 | 449389 | TAAATACACTGTTATTT | 0.822061337 | | 164 |
| 186 | chrXIV | + | 499040 | 499057 | TTTCTTTATGTTTAGCT | 0.928956769 | chrXIV | - | 499150 | 499133 | TATCTCTTCTTTTGTT | 0.820455656 | | 110 |
| 187 | chrXIV | - | 546149 | 546132 | TATTTTTACGTTTTGGC | 0.956489817 | chrXIV | + | 545981 | 545998 | AACATTAGTATTTAATT | 0.792422254 | | 168 |
| 188 | chrXIV | - | 561330 | 561313 | TTTGTTCACATTTAGTT | 0.930292374 | chrXIV | + | 561216 | 561233 | TTGATTTACATTCAAAC | 0.797477323 | | 114 |
| 189 | chrXIV | + | 609536 | 609553 | TTTTTTTATGTTTATTT | 0.986916959 | chrXIV | - | 609674 | 609657 | TATTTATGTCTTTACTT | 0.819944062 | | 138 |
| 190 | chrXIV | - | 635833 | 635816 | TTTTTTTAATTTTAGTT | 0.954915715 | chrXIV | + | 635716 | 635733 | TGTTTTTTTTTTTGCA | 0.87217818 | | 117 |
| 191 | chrXIV | - | 691680 | 691663 | GTAATTAAACATTTTGTT | 0.910156612 | chrXIV | + | 691559 | 691576 | GATATTTCCCTTTTGGA | 0.801789741 | | 121 |
| 192 | chrXV | + | 35714 | 35731 | TATATTTATATTTAGAG | 0.929297843 | chrXV | - | 35855 | 35838 | CATATTTATGTTTCATT | 0.847487414 | | 141 |
| 193 | chrXV | + | 72688 | 72705 | TTTTTTTACTTTTAGT | 0.962701666 | chrXV | - | 72794 | 72777 | TTTTATCACGTTTAGCA | 0.883721557 | | 106 |
| 194 | chrXV | - | 85366 | 85349 | TATACCTATATTTATGT | 0.817468435 | chrXV | + | 85268 | 85285 | GCTTTAATTTTTATTT | 0.887881307 | | 98 |
| 195 | chrXV | + | 113895 | 113912 | ATTGTTTATATTTTTGT | 0.943227229 | chrXV | - | 114058 | 114041 | TAATATCATGTTTTATA | 0.868893438 | | 163 |
| 196 | chrXV | + | 167003 | 167020 | TTTATTTATGTTTTCGT | 0.95396729 | chrXV | - | 167143 | 167126 | TTTAAAACTGTTTACGT | 0.78001402 | | 140 |
| 197 | chrXV | - | 277732 | 277715 | GTTGTTTATCTTTTGTT | 0.926499065 | chrXV | + | 277562 | 277579 | TTATAAAAAATTTATTT | 0.859561998 | | 170 |
| 198 | chrXV | - | 337483 | 337466 | TCTTTTTACCTTTTGTC | 0.904262836 | chrXV | + | 337385 | 337402 | TATTTTAGTATTTATTT | 0.870845988 | | 98 |
| 199 | chrXV | + | 436790 | 436807 | TATATTTATTTTTATTC | 0.935122318 | chrXV | - | 436888 | 436871 | TTCTTTTTCATTTATT | 0.832867098 | | 98 |
| 200 | chrXV | - | 490060 | 490043 | GTTGTTTTTCTTTTCTT | 0.860946443 | chrXV | + | 489890 | 489907 | TAAGTTTATATTTTGGT | 0.951016266 | | 170 |
| 201 | chrXV | - | 566597 | 566580 | AAATTTTACCTTTTGAT | 0.915947006 | chrXV | + | 566499 | 566516 | AATATTTAATATCTCTT | 0.824916747 | | 98 |
| 202 | chrXV | + | 656701 | 656718 | CTATTAATGATTAGTA | 0.901351813 | chrXV | - | 656901 | 656884 | GTTGATTTCTTTTTCTT | 0.817366446 | | 200 |
| 203 | chrXV | + | 729795 | 729812 | TATTTTTATATTTTGGC | 0.964523057 | chrXV | - | 729894 | 729877 | TTCTTTCATTTTGTAC | 0.823636542 | | 99 |
| 204 | chrXV | + | 766689 | 766706 | GTATTTTACGTTTTTTC | 0.912718329 | chrXV | - | 766791 | 766774 | TATTTAAATTTCTGTA | 0.860782306 | | 102 |

Table 2 continued on next page

Table 2 continued

| | Proposed primary ACS (Nieduszynski et al., 2006) | | | | | | Putative secondary ACS (this study) | | | | | | |
| ID | Chromosome | Strand | Start | End | Match | Score | Chromosome | Strand | Start | End | Match | Score | Protected length (nt) |
|---|---|---|---|---|---|---|---|---|---|---|---|---|---|
| 205 | chrXV | + | 783386 | 783403 | TATTTTTAACTTTTGGT | 0.942451749 | chrXV | - | 783582 | 783565 | TCTTTTTATCTCTTCAA | 0.777182413 | 196 |
| 206 | chrXV | - | 874370 | 874353 | CATTTTAATATTTGTTA | 0.881539907 | chrXV | + | 874192 | 874209 | AAGTTTTCCGTTTAGCA | 0.807155571 | 178 |
| 207 | chrXV | + | 908307 | 908324 | CTAAACTTGTTTATGT | 0.815272772 | chrXV | - | 908439 | 908422 | GGTTTTTTTTTAAGT | 0.8448056 | 132 |
| 208 | chrXV | + | 981507 | 981524 | TTTTTTATTTATATTT | 0.874148828 | chrXV | - | 981603 | 981586 | TTTTTCATGATTTTGT | 0.924378634 | 96 |
| 209 | chrXV | + | 1053687 | 1053704 | TAATTAATTGTTTTGTT | 0.896133812 | chrXV | - | 1053797 | 1053780 | CGATTAAATGTTTTTAT | 0.856030986 | 110 |
| 210 | chrXVI | - | 43150 | 43133 | TTTGTTTATATTTTTGA | 0.929263085 | chrXVI | + | 42958 | 42975 | TTCTTTTACCTTTAATA | 0.863567037 | 192 |
| 211 | chrXVI | + | 73104 | 73121 | GTTTTTTTGTTTTTC | 0.902693595 | chrXVI | - | 73301 | 73284 | TATATTTATAATTATAA | 0.896514883 | 197 |
| 212 | chrXVI | + | 116593 | 116610 | TATTTTTATGTTTTGTT | 0.998337845 | chrXVI | - | 116770 | 116753 | TAAAATTAAGTTTTGCG | 0.868507637 | 177 |
| 213 | chrXVI | + | 289531 | 289548 | ATAATTAAATGTTTACTT | 0.925413716 | chrXVI | - | 289675 | 289658 | AAAGTTAATTTTTATAT | 0.885623957 | 144 |
| 214 | chrXVI | + | 384591 | 384608 | TATTCTAAAAATTTATGT | 0.840759582 | chrXVI | - | 384718 | 384701 | TTTAAAATATATTTAAGT | 0.869580534 | 127 |
| 215 | chrXVI | + | 418177 | 418194 | TTCTTTCTTATTACAA | 0.82265266 | chrXVI | - | 418289 | 418272 | TATTATTTTGTTTTCTT | 0.900944489 | 112 |
| 216 | chrXVI | - | 456763 | 456746 | TTTTATTATTTTTGTT | 0.945433762 | chrXVI | + | 456626 | 456643 | CTTATTCACAATTCAA | 0.820656345 | 137 |
| 217 | chrXVI | + | 511708 | 511725 | TATTTTTATGTTTTTG | 0.954763972 | chrXVI | - | 511820 | 511803 | GTGGTTATCATTTATTT | 0.826572147 | 112 |
| 218 | chrXVI | + | 563881 | 563898 | AGTCTTTTATATTTAGT | 0.760925944 | chrXVI | - | 563991 | 563974 | TCTAAATATTCATCT | 0.791939697 | 110 |
| 219 | chrXVI | + | 565119 | 565136 | TGTTTTAATTTTAGT | 0.884153732 | chrXVI | - | 565272 | 565255 | TTTTTGGTCTTTGTT | 0.822137769 | 153 |
| 220 | chrXVI | + | 633925 | 633942 | CGTTTTTATAGTTTTAGT | 0.858684766 | chrXVI | - | 634064 | 634047 | TTGTTTTATATTTAACA | 0.875389458 | 139 |
| 221 | chrXVI | + | 684409 | 684426 | TTTTTTTTACTTTTTGT | 0.892233188 | chrXVI | - | 684534 | 684517 | CATATGTTTGTTTAGCT | 0.847979457 | 125 |
| 222 | chrXVI | - | 695624 | 695607 | TTTTTTTTTAATTTCT | 0.889872135 | chrXVI | + | 695470 | 695487 | AATTTTATATTTGGTT | 0.944984083 | 154 |
| 223 | chrXVI | + | 749121 | 749138 | AATTTTTAAGTTTAGTA | 0.947297384 | chrXVI | - | 749222 | 749205 | ATAATTTACATTTTATT | 0.907501113 | 101 |
| 224 | chrXVI | - | 777098 | 777081 | TTTATTTATATTTTGGC | 0.954875691 | chrXVI | + | 776923 | 776940 | AATGTGTTAGTTTTTCT | 0.811819984 | 175 |
| 225 | chrXVI | - | 819345 | 819328 | AATTTTTATATTTATTC | 0.952049491 | chrXVI | + | 819204 | 819221 | TATATTATCATATAGTT | 0.819972999 | 141 |
| 226 | chrXVI | - | 842856 | 842839 | TTTATTTAGATTTAGTT | 0.894404608 | chrXVI | + | 842714 | 842731 | AATTTTAATCTTTAGTA | 0.928064324 | 142 |
| 227 | chrXVI | + | 880904 | 880921 | CTCATATATTTTTATG | 0.822074378 | chrXVI | - | 881035 | 881018 | TAACTCTAACTTTTTA | 0.800027746 | 131 |
| 228 | chrXVI | - | 933170 | 933153 | CTTATTTACGTTTAGCT | 0.93305337 | chrXVI | + | 933047 | 933064 | ATTCAAAATATTTTGGA | 0.822210839 | 123 |

DOI: https://doi.org/10.7554/eLife.40802.009

## Acknowledgements

We thank Etienne Schwob (IGMM, Montpellier) for providing us the *orc2-1*, *orc5-1* and *cdc6-1* strains. Dirk Remus (MSKCC, New-York), Philippe Pasero (IGH, Montpellier), Armelle Lengronne and members of both Pasero and Libri laboratories for critical reading of the manuscript and fruitful discussions. Julien Soudet and Françoise Stutz (University of Geneva, Geneva) for sharing results before publication. This work was supported by the Centre National de la Recherche Scientifique (CNRS), the Fondation pour la Recherche Medicale (FRM, programme équipes 2013), l'Agence National pour la Recherche (ANR, grant ANR-16-CE12-0022-01), the Labex Who Am I? (ANR-11-LABX-0071 and Idex ANR-11-IDEX-0005–02). TC and JG were supported by fellowships from the French Ministry of Research and the Ligue Nationale contre le Cancer (allocation GB/MA/CD/IQ – 12031), respectively.

## Additional information

### Funding

| Funder | Grant reference number | Author |
|---|---|---|
| Centre National de la Recherche Scientifique | | Domenico Libri |
| Fondation pour la Recherche Médicale | F.R.M. programme équipes 2013 | Domenico Libri |
| Agence Nationale de la Recherche | Grant ANR-16-CE12-0022-01 | Domenico Libri |
| Labex | WhoamI? ANR-11-LABX-0071 | Domenico Libri |
| French Ministry of Research | Fellowship | Tito Candelli |
| Ligue Contre le Cancer | GB/MA/CD/IQ - 12031 | Julien Gros |
| Labex | WhoamI? ANR-11-IDEX-0005-02 | Domenico Libri |

The funders had no role in study design, data collection and interpretation, or the decision to submit the work for publication.

### Author contributions

Tito Candelli, Conceptualization, Data curation, Software, Formal analysis, Validation, Investigation, Methodology, Writing—review and editing; Julien Gros, Conceptualization, Data curation, Supervision, Validation, Investigation, Visualization, Methodology, Writing—original draft, Writing—review and editing; Domenico Libri, Conceptualization, Data curation, Formal analysis, Supervision, Funding acquisition, Validation, Investigation, Methodology, Writing—original draft, Project administration, Writing—review and editing

### Author ORCIDs

Tito Candelli https://orcid.org/0000-0003-2440-6032
Julien Gros http://orcid.org/0000-0002-8316-0207
Domenico Libri http://orcid.org/0000-0001-6728-0594

### Decision letter and Author response

Decision letter https://doi.org/10.7554/eLife.40802.024
Author response https://doi.org/10.7554/eLife.40802.025

## Additional files

### Supplementary files

• Supplementary file 1. Supplementary tables 1, 2, 3.

DOI: https://doi.org/10.7554/eLife.40802.014

• Transparent reporting form

DOI: https://doi.org/10.7554/eLife.40802.015

## Data availability

All data analyzed in this manuscript have been previously published and appropriate GEO accession codes and references have been provided.

The following previously published datasets were used:

| Author(s) | Year | Dataset title | Dataset URL | Database and Identifier |
|---|---|---|---|---|
| Schaughency P, Merran J, Corden JL | 2014 | Genome-wide mapping of yeast RNA polymerase II termination. | https://www.ncbi.nlm.nih.gov/geo/query/acc.cgi?acc=GSE56435 | NCBI Gene Expression Omnibus, GSE56435 |
| Candelli T, Challal D, Briand J, Boulay J, Porrua O, Colin J, Libri D | 2018 | CRAC of yeast RNA polymerase II in various thermosensitive strains at permissive and non-permissive temperature and anchor-away strains with the addition of rapamycin. | https://www.ncbi.nlm.nih.gov/geo/query/acc.cgi?acc=GSE97913 | NCBI Gene Expression Omnibus, GSE97913 |
| Roy K, Gabunilas J, Gillespie A, Ngo D, Chanfreau GF | 2016 | 3′-end sequencing of poly(A)+ RNA in wild-type Saccharomyces cerevisiae and nuclear exosome mutant strains | https://www.ncbi.nlm.nih.gov/geo/query/acc.cgi?acc=GSE75586 | NCBI Gene Expression Omnibus, GSE75586 |

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
