## [Decision Letter]

Thank you for submitting your article "Pervasive transcription fine-tunes replication origin activity" for consideration by *eLife*. Your article has been reviewed by three peer reviewers, and the evaluation has been overseen by Bruce Stillman as Reviewing Editor and Kevin Struhl as the Senior Editor. The reviewers have opted to remain anonymous.

The reviewers have discussed the reviews with one another and the Reviewing Editor has drafted this decision to help you prepare a revised submission.

Summary:

This paper aims to define how transcription influences replication origin function in budding yeast. It is known that transcription through origin sequences can influence DNA replication, but it is unclear how many origins are affected. mRNA abundance, measured at steady state, is a poor indication of ongoing transcription so Candelli et al. use a newer method to map nascent RNA and RNA polymerase occupancy across the genome. Their most prominent conclusions are: (1) that replication origins function as strong transcription terminators; (2) termination sites are correlated with ACS locations; (3) most origins utilize two ACS sequences – by implication two sites for ORC loading; (4) ORC, Cdc6 or the entire pre-RC mediates transcription termination; (5) transcription through some replication origins influences origin activity.

Taken together this report provides a solid advance for our understanding of how transcription may influence origin activity, or how assembly of proteins at origins of DNA replication affect transcription. The major finding, which is convincingly supported by the data, is the demonstration that proteins assembled in an ORC- and Cdc6-dependent manner can inhibit transcriptional elongation. This result is likely to be of broad interest. The second important finding is that the level of transcription across origins may influence origin activity. Beyond these findings the paper does not elaborate a more comprehensive understanding of how transcription may influence origin function. The role of the second ACS/ORC binding site is not particularly well supported; how and why some origins are more sensitive to transcription is also not defined.

The paper is of sufficient interest and the authors are requested to respond to specific issues, some relating to individual conclusions. The following are specific issues that the reviewers have raised about the manuscript that need to be addressed before publication.

Essential revisions:

1) It is clear that transcription can terminate in the vicinity of the primary ACS, however the data showing termination near the secondary ACS is unconvincing. The data in Figure 1D, E are noisy (see point 2 below) and the small peak near the secondary ACS could be an artifact from a single over-represented site.Further investigation of the second ACS site in the paper is also not particularly convincing. Thus, the conclusions about these data supporting the model that multiple ORC proteins assemble the pre-RC are not warranted.

2) Related to point 1, the authors use the technique of Schaughency et al., 2014 for measuring RNA reads at genomic loci. The data in Schaughency et al., 2014 show mean reads of 30-60 (e.g., near Poly A sites), but the data in Figure 1 A show mean number of RNAPII reads of 2-7 near origin sequences. How significant are these reads? Is there anywhere in the genome that does not have reads of at least 2-7 (i.e., lacks RNAPII)? The low reads could be experimental background and the dip could be because a protein (ORC-Cdc6) is bound at the origin. Again, what are the mean reads near poly(A)+ sites analyzed in Figure 1B. Only a summary of the location of these reads is shown. Showing the mean Poly(A)+ reads over the origin in addition to the summary analysis would be more convincing. Figure 1D: Again, the number of reads is very low and this could be due to random noise. What is convincing about the peaks, particularly at secondary ACS sites?

3) The data in Figure 2 is interpreted to show significant roadblock of RNA polymerase II at ACS sites. However, given that the mapped signal fluctuates significantly across the regions shown, it is unclear how the authors can conclude that one specific site is a "roadblock" whereas another nearby site isn't? Specifically, Figure 2A and subsection “RNAPII pausing and transcription termination occur at ARS borders”. The light green track in Figure 2A show that in the mutant, many peaks of RNAPII pausing are observed, but the authors only point to the ACS. Why? What about all the other peaks in the track? Same for Figure 2B, Figure 2C and Figure 2D and yet the authors point out that ACSs are occupied by ORC. But ORC or the pre-RC is most likely not at the other pauses.

4) The Abstract states, " We provide evidence that quasi-symmetrical binding of the ORC complex to ARS borders is responsible for pausing/termination." This is too strong a statement since they have not shown that ORC does this. Indeed, it is shown that Cdc6 mutants also compromise termination. It could be that loaded pre-RC or the Mcm2-7 double hexamers are the terminator, not ORC. Please clarify the abstract, otherwise it will become accepted that ORC causes termination

[Editors' note: further revisions were requested prior to acceptance, as described below.]

Thank you for resubmitting your work entitled "Pervasive Transcription Fine-tunes Replication Origin Activity" for further consideration at *eLife*. Your revised article has been favorably evaluated by Kevin Struhl (Senior Editor) and a Reviewing Editor.

The revised paper has been improved and the authors have addressed the questions raised by the reviewers. As such it is now appropriate for publication in *eLife*, with some modifications or a brief response to issues raised below. The paper does show that proteins assembled at ARS origins can contribute to termination of transcription through the origin. The problem remains in the interpretation of the results and here the authors even propose contradictory ideas. In re-reading the paper, there are some miss-statements or contradictions that could be corrected.

These are:

Introduction, the authors should reference that Steve Bell's laboratory, who performed the cited single molecule experiments, but has published subsequent research consistent with the two ORC-Cdc6 model. Thus, the paragraph about a controversy should be toned down and their reference cited in the Introduction [Warner et al. (2017)].

Introduction. The paragraph starts of by stating that "Studies have proposed that transcription might activate replication origins" and then in the second sentence Marahrens and Stillman, 1992 is cited as evidence. Marahrens and Stillman, 1992 did not propose that transcriptional activators activate replication by via transcription, but by activators likely modulating chromatin structure. Thus, this paper does not support the hypothesis by the authors. Even Stagljar et al., 1999 did not show that transcription activates an origin. These papers are not "in apparent contrast with the demonstration that strong transcription through ARSs is detrimental for their function" (Introduction). While it is true that transcription is detrimental for replication initiation, accurate citing of references is needed.

Subsection “Topological organization of replication origin factors detected by transcriptional footprinting” and later in subsection “Functional implications for pervasive transcription at ARS”. The authors argue that transcription may "play an important role" in the initiation of DNA replication by pushing the first loaded Mcm2-7 hexamer away from the ORC binding site. This is clearly not the case since initiation of DNA replication in vitro is very efficient and does not require transcription, even on chromatin templates (Kurat et al., 2017). It seems the authors are stating on one case that ARSs and their binding proteins terminate transcription and yet then state that transcription may play an important role in replication initiation. This, based on the data presented, does not seem likely (at least it has not been demonstrated). Indeed, subsection “Functional implications for pervasive transcription at ARS”, the authors state " We propose that transcription though origins might induce similar changes that are susceptible to outcompete binding of ORC and/or pre-RC formation." They cannot have it both ways.

---

## [Author Response]

Essential revisions:1) It is clear that transcription can terminate in the vicinity of the primary ACS, however the data showing termination near the secondary ACS is unconvincing. The data in Figure 1D, E are noisy (see point 2 below) and the small peak near the secondary ACS could be an artifact from a single over-represented site. Further investigation of the second ACS site in the paper is also not particularly convincing. Thus, the conclusions about these data supporting the model that multiple ORC proteins assemble the pre-RC are not warranted.2) Related to point 1, the authors use the technique of Schaughency et al., 2014 for measuring RNA reads at genomic loci. The data in Schaughency et al., 2014 show mean reads of 30-60 (e.g., near Poly A sites), but the data in Figure 1 A show mean number of RNAPII reads of 2-7 near origin sequences. How significant are these reads? Is there anywhere in the genome that does not have reads of at least 2-7 (i.e., lacks RNAPII)? The low reads could be experimental background and the dip could be because a protein (ORC-Cdc6) is bound at the origin. Again, what are the mean reads near poly(A)+ sites analyzed in Figure 1B. Only a summary of the location of these reads is shown. Showing the mean Poly(A)+ reads over the origin in addition to the summary analysis would be more convincing. Figure 1D: Again, the number of reads is very low and this could be due to random noise. What is convincing about the peaks, particularly at secondary ACS sites?

Essential revisions #1 and #2 are very much related and will be answered together below. For matching questions and answers we added to each question or group of related questions a letter (A, B and C) that refers to the answer.

A) On the significance of RNAPII pausing at the primary ACSs

The first concern we will discuss is the significance of the distribution of RNAPII around origins at the primary sites. As stated in the manuscript, the main aim of these analyses was to assess the impact of the low levels of pervasive transcription around origins, transcription that is generally non-annotated and often due to readthrough at canonical terminators. It is therefore expected that the average level of reads around origins be lower than the genome average of 30-60 reads, since we are sampling the lowest percentile of the distribution. Importantly, however, we did not profile the *mean* levels of reads but the *median*, which was done precisely in order to undermine the contribution of highly represented sites. With a less stringent analysis, i.e. when plotting the mean values (Author response image 1), the levels of the RNAPII reads in the region of the roadblock (around 15) are very comparable with the genome average as cited by the referee (30-60) and the drop in the signal clearly visible.

**Author response image 1. respfig1:** RNAPII ParCLIP reads (mean values) are profiled around origins aligned on the first nucleotide of the primary ACS.

However, we believe that presenting these data could be misleading, as at least a fraction of the signal at the roadblock could be due to a very limited number of sites with high values. Indeed, some peaks are only visible using the mean (see for instance the peak at +200) and clearly due to outliers that do not represent the overall population (in this case probably a site of initiation after the origin).

Use of the median is more stringent and generally more appropriate for representing distribution that deviate from normality.

We considered that good evidence for the existence of a significant signal at the roadblock would be loss of that signal immediately after the ACS. Therefore, we compared the reads levels in the 100nt before and after the ACS for every region (Figure 1—figure supplement 2A). The distributions of these values for the 100 origins with the highest levels of surrounding transcription are now shown in Figure 1—figure supplement 2B under the form of Box Plots. It appears clearly that the median signal is higher before the ACS and drops immediately after, and that the loss of signal is highly significant according to both parametric and non-parametric tests. A strong statistical significance is observed even when all the origins are considered (data not shown).

This signal derives from the crosslinking of the nascent RNA to the polymerase, and the absence of signal (and nascent RNA) can only derive from the failure of RNAPII from actively transcribing that particular region. The drop in the RNAPII signal occurs thereforebecause of the presence of a protein complex bound at the ACS. This is not a technical artefact as the one that could be expected from ChIP datasets, in which the absence of crosslinking could be due to the steric hindrance due to another complex bound at the same location. Lastly, we would also like to stress that the poor signal around ACSs cannot be ascribed to the poor "mappability" of reads derived from such AT rich regions, because: i) similar AT-rich regions elsewhere in the genome have signals and ii) a signal at origins can be detected when incoming transcription is "forced" in transcription termination mutants (e.g. *rna15-2* or *NRD1::AID*).

B) On the significance of RNAPII pausing at the secondary ACSs

We agree with the referees that we did not provide a strong experimental support for the existence of a second ORC complex bound to the secondary ACS. We did our best to tone down these claims and we only claim consistency with this hypothesis in the revised manuscript.

Concerning the analyses of transcription around the putative secondary ACSs, we did not intend to claim that secondary ACSs induce transcription termination and apologize if this was not sufficiently clear in the manuscript. In Results section and subsection “Topological organization of replication origin factors detected by transcriptional footprinting”of the original manuscript we had proposed that the best interpretation of the data shown in Figure 1D-E is that RNAPII pauses at the secondary ACS but that termination occurs later on, which is actually the basis of the asymmetry that we observed. We have added a few sentences to strengthen this notion in the revised version of the manuscript. We also added a probability profile of termination around secondary ACS (see below, Figure 1—figure supplement 2E) from which it is clear that statistically significant termination only occurs after the ACS.

Concerning the significance of the RNAPII occupancy peak upstream of the putative secondary ACSs, we plotted in Figure 1D the *median* number of reads and not the average. By definition, the peak in the median profile upstream of the secondary ACSs cannot be due to the contribution of only a single (or a few) overrepresented values as it depends on half of the values of the distribution. To further support the significance of RNAPII pausing upstream of the secondary ACSs, we compared the distributions of RNAPII levels before and after the aligned, putative secondary ACSs. Here again we found a very significant decrease in the signal (Figure 1—figure supplement 2C).

C) On the significance of termination at primary ACSs

Finally, the referees request the metaprofile of the mean level of RNA 3’-ends around ACSs. We would like to stress that the question addressed in Figures 1B,1E, Figure 4A and 4B was whether termination occurred upstream of origins, using as a proxy the presence/absence of RNA 3’ ends in the regions analyzed. Because the RNAs produced can have different stabilities, the average 3’-ends signal (as opposed by the 3’-ends count) is strongly influenced by the steady state level of the RNAs. Using this indicator for termination might introduce a major bias, as one or a few RNAs with high steady state levels would dominate the signal, which would be artefactual. This was particularly relevant at origins because many of the RNAs produced in these regions are poorly abundant, and because roadblocked transcription events tend to produce mainly non-coding and unstable RNAs (Colin et al., 2014).

Since it is the occurrence of termination events that we profile independently of the RNA steady state levels, profiling the average 3’-ends signal would therefore not be appropriate.

Nevertheless, to convince the referees that there is increased occurrence of termination events immediately preceding the average ACS, we calculated the statistical significance of the observed number of termination events at the ACS peak. To do so, we adopted the H0 hypothesis that termination occurs with equal frequency in the whole region of alignment (-500 to +500 from the ACS), and calculated a p-value for each position based on the frequency observed in the first 100nt window (position -500) and on the actual values observed at every position.

As shown in Figure 1—figure supplement 2D, the frequency of termination events is not significantly different in most of the region. However, a prominent peak of very low p-value is seen immediately *upstream* of the ACS, demonstrating that termination occurs with higher frequency in this region (p<10^-20^). We also performed the same analysis around secondary ACSs (Figure 1—figure supplement 2E), from which it is clear that termination occurs with high significance only *after* the ACS.

We conclude from these analyses, which have been included in the revised version of the manuscript, that transcription termination occurs immediately upstream of the primary ACSs with high statistical significance.

3) The data in Figure 2 is interpreted to show significant roadblock of RNA polymerase II at ACS sites. However, given that the mapped signal fluctuates significantly across the regions shown, it is unclear how the authors can conclude that one specific site is a "roadblock" whereas another nearby site isn't? Specifically, Figure 2A and subsection “RNAPII pausing and transcription termination occur at ARS borders”. The light green track in Figure 2A show that in the mutant, many peaks of RNAPII pausing are observed, but the authors only point to the ACS. Why? What about all the other peaks in the track? Same for Figure 2B, Figure 2C and Figure 2D and yet the authors point out that ACSs are occupied by ORC. But ORC or the pre-RC is most likely not at the other pauses.

We thank the referees for giving us the opportunity to clarify this point. The snapshots in Figure 2 are only shown with the purpose of illustrating the behavior of RNAPII around specific origins. These snapshots alone do not demonstrate that the pausing that is observed is specifically due to the presence of an origin. Indeed, many additional sites of pausing are observed within genes and sometimes in the downstream region, at a distance from origins. What is telling us that ACS sequences induce pausing is the aggregate signal (Figure 1), in which other sites of pausing are averaged out while pausing immediately preceding the ACSs remains visible. This implies that pausing occurs at the *majority* of origins; otherwise it would not be detected by the *median*. Also, note that RNAPII pausing peaks at ACSs often appear after a region of low signal, which is consistent with an accumulation of polymerases fed by low levels of upstream readthrough transcription. To better highlight this point we have modified Figure 2 by adding an inset at panel 2A.

4) The Abstract states " We provide evidence that quasi-symmetrical binding of the ORC complex to ARS borders is responsible for pausing/termination." This is too strong a statement since they have not shown that ORC does this. Indeed, it is shown that Cdc6 mutants also compromise termination. It could be that loaded pre-RC or the Mcm2-7 double hexamers are the terminator, not ORC. Please clarify the abstract, otherwise it will become accepted that ORC causes termination.

We agree, we did not show that ORC alone is sufficient for termination, only that is necessary. We modified the abstract as requested.

[Editors' note: further revisions were requested prior to acceptance, as described below.]

The revised paper has been improved and the authors have addressed the questions raised by the reviewers. As such it is now appropriate for publication in eLife, with some modifications or a brief response to issues raised below. The paper does show that proteins assembled at ARS origins can contribute to termination of transcription through the origin. The problem remains in the interpretation of the results and here the authors even propose contradictory ideas. In re-reading the paper, there are some miss-statements or contradictions that could be corrected.

We thank the editors for re-evaluating favorably our revised manuscript. We briefly answer below to the last concerns.

These are:Introduction, the authors should reference that Steve Bell's laboratory, who performed the cited single molecule experiments, but has published subsequent research consistent with the two ORC-Cdc6 model. Thus, the paragraph about a controversy should be toned down and their reference cited in the Introduction [Warner et al. (2017)].

This reference had been included in the original manuscript subsection “Topological organization of replication origin factors detected by transcriptional footprinting”) to discuss the possible sliding of an intermediate during helicase loading. Results presented in Warner et al. are consistent with the quasi-symmetrical model, as the authors suggest, but do not prove that ORC binds to the B2 element when sliding is prevented. We therefore cited the work as "see also", together with the Coster et al. article. We also eliminated the notion of controversy, but we believe that the single molecule studies of the Bell and Greene laboratories should still be referenced.

Introduction. The paragraph starts of by stating that "Studies have proposed that transcription might activate replication origins" and then in the second sentence Marahrens and Stillman, 1992 is cited as evidence. Marahrens and Stillman, 1992 did not propose that transcriptional activators activate replication by via transcription, but by activators likely modulating chromatin structure. Thus, this paper does not support the hypothesis by the authors. Even Stagljar et al., 1999 did not show that transcription activates an origin. These papers are not "in apparent contrast with the demonstration that strong transcription through ARSs is detrimental for their function" (Introduction). While it is true that transcription is detrimental for replication initiation, accurate citing of references is needed.

We had cited these papers for reporting that transcription activators binding is required for efficient origin firing (Introduction: "The binding of general transcription factors such as Abf1 and Rap1, or even the tethering of transcription activation domains, TBP or Mediator components was shown to be required for efficient firing of a model ARS"). Indeed, these studies do not show that transcription is induced at the studied origins, but they do not prove either that it is not. In the Stagljar et al. paper, this is actually suggested. This is why we considered this in apparent contradiction with the notion that strong transcription inactivates origins. We clarified this point and also deleted the first sentence that was associated to incorrect references. The reference to the Knott et al. study was associated to the references describing the importance of transcription factors at origins.

*Subsection “Topological organization of replication origin factors detected by transcriptional footprinting” and later in subsection “Functional implications for pervasive transcription at ARS”. The authors argue that transcription may "play an important role" in the initiation of DNA replication by pushing the first loaded Mcm2-7 hexamer away from the ORC binding site. This is clearly not the case since initiation of DNA replication* in vitro *is very efficient and does not require transcription, even on chromatin templates (Kurat et al., 2017). It seems the authors are stating on one case that ARSs and their binding proteins terminate transcription and yet then state that transcription may play an important role in replication initiation. This, based on the data presented, does not seem likely (at least it has not been demonstrated). Indeed, subsection “Functional implications for pervasive transcription at ARS”, the authors state " We propose that transcription though origins might induce similar changes that are susceptible to outcompete binding of ORC and/or pre-RC formation." They cannot have it both ways.*

Our analyses are based on the average, negative effect of pervasive transcription on replication initiation. They do not exclude that in individual cases, pushing of the DH by transcription might also favor initiation of replication (even though this is not required in vitro). This is the reason of the apparent contradiction. Nevertheless, we agree that this is only a point of discussion and that we did not address here this possibility. Therefore, we eliminated the claim that transcription might favor firing by deleting this sentence.

Lastly, we also added a missing reference to the recent Soudet et al. paper.